# Physics of Space Weather Phenomena: A Review

**Ashok Kumar Singh [1,\*], Asheesh Bhargawa [1], Devendraa Siingh [2] and Ram Pal Singh [3]**

1 Department of Physics, University of Lucknow, Lucknow 226007, India; asheeshbhargawa@gmail.com
2 Indian Institute of Tropical Meteorology, Pune 411008, India; devendraasiingh@gmail.com
3 Department of Physics, Banaras Hindu University, Varanasi 221005, India; singhrp_03@yahoo.co.in
\* Correspondence: aksphys@gmail.com

**Abstract:** In the last few decades, solar activity has been diminishing, and so space weather studies need to be revisited with more attention. The physical processes involved in dealing with various space weather parameters have presented a challenge to the scientific community, with a threat of having a serious impact on modern society and humankind. In the present paper, we have reviewed various aspects of space weather and its present understanding. The Sun and the Earth are the two major elements of space weather, so the solar and the terrestrial perspectives are discussed in detail. A variety of space weather effects and their societal as well as anthropogenic aspects are discussed. The impact of space weather on the terrestrial climate is discussed briefly. A few tools (models) to explain the dynamical space environment and its effects, incorporating real-time data for forecasting space weather, are also summarized. The physical relation of the Earth's changing climate with various long-term changes in the space environment have provided clues to the short-term/long-term changes. A summary and some unanswered questions are presented in the final section.

**Keywords:** space weather; solar activity; solar wind; solar flares; coronal mass ejections

## 1. Introduction

Space weather, the highly dynamic and variable conditions in space, is impacted by changing solar radiations due to varying conditions of plasma, magnetic field, highly energetic charged particles, etc. [1–5]. Variations in these sources causes a difference in the ambient conditions of the terrestrial atmosphere, ionosphere, magnetosphere and interplanetary space, which house the entire modern communication system, transport system, energy system, and even military movements and command systems, etc. The modern equipment used in all these systems are designed to operate on average, steady-state properties of the environments, with allowance for some variations. Large variations in the environmental conditions may lead to malfunctioning/non-operational conditions, and hence, potential impact on the technological/biological systems [6–8]. Schrijver et al. [9] presented a global road map on the understanding of space weather to shield society from damage.

The Sun, a G-type main sequence star [10] and a crucial source of energy for all living systems on the Earth, is the epicentre for variable space weather conditions. The major constituents of the Sun are hydrogen (~73%) and helium (~25%), with some other elements such as iron, carbon, neon and oxygen, etc. contributing by ~2% [11]. The Sun's activity, which is measured by intensities of electromagnetic and charged particle radiation, exhibits a cyclical variation of a period of 11 years, and during this time span, the intensity of radiations varies from minimum to maximum and then again goes to minimum [12–17]. The Sun also shows periodic changes during the evolution of irregular structures at its surface [18]. These structures are basically the sunspots associated with strong magnetic fields and are surrounded by active regions within which they could grow [19].

The interior structure of the Sun is divided into different layers/zones, where the inner most zone (core) acts as a thermonuclear reactor and produces most of the Sun's energy

through the fusion of hydrogen into helium atoms. The thermal energy generated into the core propagates outward into the radiation zone, convective zone and photosphere, and ultimately, the temperature reduces from core ($1.5 \times 10^7$ K) to photosphere (5778 K) [20,21]. This high temperature difference results in a pressure gradient between the corona and the photosphere and provides enough velocity to ionized particles at the corona, and further allows particles to escape from the Sun's atmosphere [22,23].

There is considerable scope for future research that advances our ability to deliver accurate and useful space weather services. However, the momentum of that research will only be sustained if we maintain and improve wider knowledge of space weather risks, not least by ensuring that policymakers and the general public are aware of research progress and how it can be exploited to protect societies and economies around the world. This is not a job that can be carried out once, as there are always more people with whom the scientific community can engage. In particular, there is a steady turnover of people in decision-making positions, and we need to raise their awareness so that we can promote evidence-based policy on space weather, both in government and industry. This engagement is a vital element in supporting future research, not least to emphasize the scientific evidence on the real risk from severe space weather conditions.

There is a need to understand the dynamic variability of the Sun or the space weather, which motivates the solar terrestrial physics community to think of a more integrated view of the entire chain of processes involved in the Sun, in the interplanetary spaces and in the magnetosphere ionosphere system. To know about the better understanding of the space weather, we have to deal with the solar as well as terrestrial perspectives and also the other (anthropogenic) aspects. This review paper is suitably divided into eight sections. Section 1 presents the customary introduction. Section 2 is devoted to the solar perspectives of space weather, while terrestrial perspectives are presented in Section 3. Section 4 describes the predictability and future prospects of space weather phenomena. Section 5 briefly discusses the role of satellites/space-based observations in space weather studies. Section 6 presents the discussion, while a brief summary is provided in Section 7. At last, some unanswered questions are presented in Section 8.

## 2. Space Weather: The Solar Perspective

Solar wind emitted from the solar surface consists of corpuscular radiation and a magnetic field, and is the most effective driver of space weather activity. The high-speed solar wind stream interacting with the Earth's magnetosphere drives geomagnetic activity [24,25], may accelerate energetic electrons in the radiation belt [26,27] and may even enhance ultra-low frequency (ULF) fluctuations in the magnetosphere [28]. The sign of the IMF BZ and solar wind are also the known drivers of CMEs and ICMEs [29,30].

X-ray and UV emissions during solar flares cause increased ionization at lower altitudes (60–90 km) that absorbs high frequency (3–30 MHz) radio signals heading to radio blackouts [31]. Solar energetic particles accelerated by flares and CMEs cause biological effects and also impact electronic devices based on the ground as well as in space [32]. CMEs directed towards the Earth and interacting with the geomagnetic field produce geomagnetic storms which affects the power grid, degradation of satellite navigation, spacecraft, high-frequency communication systems, etc [33,34]. In space weather impact assessment, geomagnetic storms are considered to be a primary component [8,35,36].

### 2.1. Solar Activity

Solar activity exhibits several processes, including magnetic fields, flares, prominences, stellar winds, etc., but the magnetic field seems to be the main player in solar activities. Precisely, plasma beta (β) (the ratio of the plasma pressure to the magnetic pressure) plays the decisive role [37]. The topological properties of magnetic fields are also directly/indirectly associated with sunspots, flares, CMEs, solar wind acceleration and coronal heating [38]. The magnetic field originating in a dynamo action inside the Sun shows 11-year periodicity with maximum and minimum activities. The pattern is usually referred to as the sunspot

cycle, the solar cycle or the solar activity cycle. Figure 1 shows the variation of F10.7 solar flux and sunspot numbers, the two major parameters in the study of solar activity for last four (roughly) solar cycles. The F10.7 solar flux and the sunspot numbers almost show similar variability trends, with maxima/minima occurring at same time. This is because of the fact that the sunspot number represents magnetic activity on the Sun which controls solar flux. The transition pattern from the maximum to the minimum in the last solar cycle shows a slow rate as compared to the three previous cycles. The maximum value of solar flux/sunspot numbers in the present cycle is quite low [15,16]. This may be associated with the significant decrement in solar magnetic energy density (SMED) observed during the past four solar cycles [39].

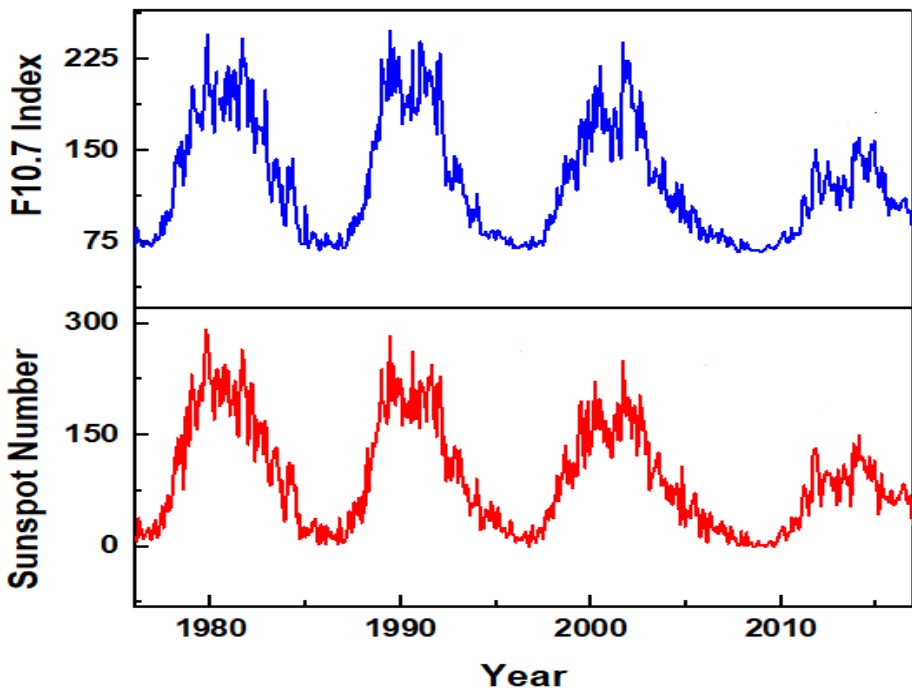

**Figure 1.** Variation of 10.7 flux and sunspot number for last three solar cycles [15].

Solar activity and fluctuations in associated parameters influence both the interplanetary space as well as the geospace. As a result, a chain of processes occurs, whose impact on the Earth's surface is observed and is depicted in Figure 2. To reduce/mitigate the dangerous impact, physical understanding of the chain of processes is essential, which includes a combination of observations, data analysis, interpretations and theoretical/empirical modelling. In the observational domain, continuous data from the Sun and inner heliosphere from a fleet of spacecraft, including the Solar Terrestrial Relations Observatory Ahead/Behind (STEREO A/B), the Solar Dynamics Observatory (SDO), the Solar and Heliospheric Observatory (SOHO), the Mercury Surface Space Environment Geochemistry and Ranging (MESSENGER) spacecraft, Venus Express (VEX), the Advance Composition Explorer (ACE), and Wind are available. The availability of refined/detailed observations made it possible to develop global magnetohydrodynamics (MHD) numerical simulations and have greatly improved our understanding of solar transients and their impact on the Earth [40–42].

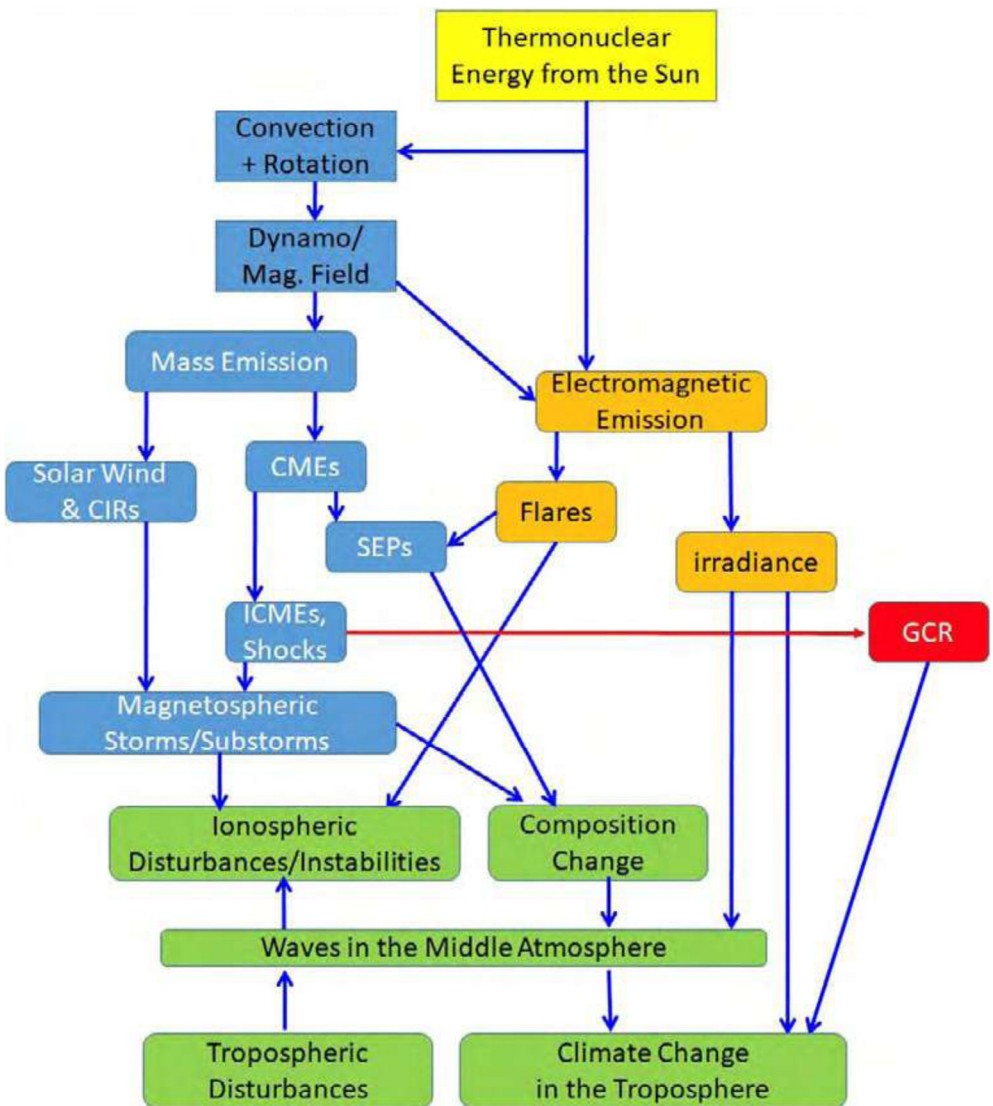

**Figure 2.** Chain of various processes in the Sun–Earth system (available online: www.issibj.ac.cn/Publications/Forum_Reports/201404/W020200522380996345108.pdf) (accessed on 10 March 2020).

### 2.1.1. The Solar Wind

Solar wind, divided into the slow (250–450 km s$^{-1}$) and the fast (450–900 km s$^{-1}$) category, seems to be bimodal, with distinct source regions, e.g., fast from coronal holes. At 1 AU, the electron number density of the slow wind is typically 5–20 electrons per cm$^3$. The positive charges are mostly protons (∼95%) with small fractions of helium nuclei (∼4%) and heavier elements (<1%). The electron temperature of the slow wind is typically 103 K and proton temperature $3 \times 104$ K, whereas the fast wind is more tenuous, some 3 electrons per cm$^3$, and much hotter, with the electron temperature typically being 105 K, and the proton temperature $2 \times 105$ K [43].

The large variability in density and temperature raises doubts about the bimodal distributions of solar wind. In fact, fast wind emerges from coronal holes, while the source of slow wind is not well understood [14,44,45]. The variability of solar wind parameters may follow some kind of dynamic continuum. However, the Ulysses mission demonstrated the out-of-ecliptic bimodality of the solar wind, which is not as evident in ecliptic observations. The Ulysses and ACE measurements of elemental abundances and ionisation states of oxygen and carbon elements are successfully used to distinguish fast and slow wind, as compared to flow speed [46]. However, the main question about the origin of fast and slow wind streams remains unanswered.

A stream of high-speed solar wind may interact with preceding slower solar wind or form stream interaction regions (SIRs). Both the fast wind and the SIRs, often called corotating interaction regions (CIRs), are important drivers of weak and moderate magnetospheric storms. The fast and slow winds escape radially from adjacent regions. The Parker spiral formed by solar rotation may lead to the fast forward compressional wave by pushing fast wind against slow wind. If the slow wind lags behind the fast wind, then reverse reflection wave may be generated [47,48].

The Source Regions of Solar Wind Streams

The source region of fast and slow wind can be identified by the in situ measurement of elemental abundances which have different values in different structures. The abundances in fast wind and slow wind are different. It seems easy to compare abundances, but due to the larger variability in it, even for the same kind of structures, it becomes very difficult to obtain a clear picture of the type of wind stream.

(1) Slow Wind Sources

Streamers have long been considered to be the source of slow wind, although the origin of outflows is not well defined. Streamers are closed loop structures that open up at their cusp, and it may be considered that some plasma may escape from them, showing tips. This is corroborated from the white blobs that flow outwards along streamers' stalks. Considering the blobs to be the tracers of slow wind and examining their outward trajectories, it is reported that slow wind may originate from about 3 to 4 solar radii [49,50]. However, it is not clear whether the slow wind originates within the streamer core or along the boundaries of the streamers, or in the streamers' stalks. The inner core configuration of a streamer is not well understood. For example, the core may be constituted of magnetically closed substructures intermixed with open field lines, and slow wind might flow along these open lines. As a result, depletion in heavy ion emission will be observed. The other possibility is reconnection between the highly stretched magnetic field lines in the streamers' stalks, which may result in the formation of plasmoids, observed as blobs. Magnetic reconnection is also likely to occur between the outer closed field lines at the streamer boundaries and the adjacent open field lines, leading to flows along the newly created open field. In addition, a streamer 'evaporation' may occur when magnetic pressure is high enough to overcome magnetic tension stretching field lines to infinity, creating new open flux [51,52]. Odstrcil et al. [53] presented a coronal structure quite similar to streamers, termed as pseudo streamers. These pseudo streamers originated in the unipolar area between coronal holes of same polarity. In such a case, no current sheath is embedded in their stalks, and as such, a wind production mechanism is ruled out, which invoked reconnection within the stalk of the structure. Still, based on ACE data analysis, pseudo streamers are reported to be the source of slow wind [54], although the mechanism of wind origin is not discussed.

Based on the analysis of interplanetary scintillation data, it was considered that the active regions on the Sun may be the source of solar wind [55]. The abundances in the wind steam video outflows from the edges of active regions are the same, and hence, association is proposed. An intermittent and continuous flow from active regions has been observed, but it is not clear whether slow wind is persistent or sporadic from the active region. Sometimes, outflows may not have sufficient speed to escape outwards. The cause of outflow needs to be explored. It has been indicated that lateral expansion of the active region may be responsible for plasma acceleration [56]. Sakao et al. [57] reported that almost one fourth of the total mass loss rate of wind could be attributed to active regions. In brief, wind from low-latitude small coronal holes is recognized to be slower than wind from large polar holes. This is in agreement with a prediction for the inverse relationship between expansion factors and wind speed.

(2) Fast Wind Sources

Coronal holes are considered to be the source of fast wind. In that case, a question that remains to be answered is whether the whole area of coronal holes evenly contributes to solar wind emission or whether there are sites within coronal holes that are preferential wind sources, such as bright points (BPs), plumes and transient X-ray jets. Bright points considered as mini active regions are easily seen in X-ray images of coronal holes. Bright points are considered as a source region because the estimation of particle flux in the region is found to be sufficient to maintain flux observed at 1 AU ($2 \times 10^8$ cm$^{-2}$ s$^{-1}$). Another indirect support is the anti-correlation between bright points and sunspot cycle. In contrast, the changes in bright point number density do not show any correlation with changes in coronal hole mass flux. Additionally, the density of bright points, taking into account variations in background corona, is found to be independent of the solar cycle. This is not in favour of bright points being the source of solar wind. Analysis of HINODE and TRACE data showed evidence for an unresolved structure in bright points [58]. Considering bright points to release their entire mass content to the solar wind to compensate its mass content, the lifetime of bright points comes to be ~100 sec, which is too small when compared to the duration of bright points, observed to be few hours [59]. Therefore, it seems that bright points are not significant contributor to solar wind.

Another possible site of solar wind origin is plumes [60], which are observed in distant solar wind. Different wind features, such as pressure-balanced structures, micro streams and switchbacks, have been proposed as in situ signatures of plume remnants. However, no definite association with coronal plasma has been established [55,61]. Numerous X-ray jets observed in plumes in HINODE data support the association of plumes to solar wind. However, it is not yet clear whether all jets escaping the gravitational attraction of the Sun will feed the wind. Rough estimation suggests that about 1/10 of wind mass flux may be supported by plumes [62]. Therefore, further observations and theoretical works are needed to ascertain the source location of winds. In fact, the observed latitudinal changes of the solar wind structure, over the solar cycle, may be a consequence of the spread of activity to higher latitudes, as the solar cycle progresses towards its maximum and different activity phenomena, which occur at an increasingly higher frequency.

### 2.1.2. CMEs and ICMEs

The ICME is either accelerated or dragged by the solar wind on which the CME is embedded, depending upon the relative velocity of CMEs. CMEs with speed slower than the solar wind velocity are accelerated when the structure (solar wind + ICME) hits the Earth's magnetosphere. The resultant magnetic field may have a large southward component, leading to a magnetic storm. CME's speed and its orientation usually determine the strength of eruption of storms. The Ampere force J×B arising from a rapid release of magnetic tension in the erupting structure causes acceleration. The famous 23 July 2012 event left the Sun with a speed of $3050 \pm 260$ km s$^{-1}$ and had the speed of $2246 \pm 110$ km s$^{-1}$ at 1 AU [63] and caused an extreme magnetic compression, with the magnetic field exceeding 100 nT. In this case, two CMEs from the same region one after the other originated and propagated outwards. It seems that the first CME creates an easier path for the following CME, which catches the leading CME, and the two structures are merged. The combined, complicated structure leads to a very strong event [64].

Further details of the large variety of the interactions of ICMEs with their surroundings may be found in Kilpua et al. [65]. It is important to note that there are different causes for the strong ICME-related southward component of the interplanetary magnetic field BS. The flux ropes with strong helical magnetic fields usually develop in the coronal region, whereas BS accumulates during propagation through the sheath region. However, further understanding of ICMEs can be developed with detailed observations.

(1) The Physical Processes of CME/ICME Evolution

After the first observation of CMEs from a coronagraph on board NASA's Seventh Orbiting Solar Observatory (OSO-7) on 14 December 1971 [66], the Coronagraph/Polarimeter

on the Solar Maximum Mission (SMM) [67] and the Large Angle Spectrometric Corona-graph (LASCO) onboard the Solar and Heliosphere Observatory (SOHO) [68] have reported thousands of CMEs and their characteristics features. Gopalaswamy [11] discussed the structure of interplanetary CMEs (ICMEs). CMEs are transients that are observed by the classical coronagraphs (LASCO C3) up to 30 R$\odot$ (solar radii) and transients beyond this range are considered ICMEs.

The launch of the twin Solar Terrestrial Relations Observatory (STEREO) space-craft [69] provides continuous observations of CMEs including the plasma and magnetic field from the Sun to the Earth [70,71]. In addition, the availability of super computers has made it possible to study the CME events by three-dimensional magnetohydrodymanics (MHD) simulations [53,72–74]. Webb and Howard [75], using STEREO and other spacecraft data, summarised observational features of CMEs, and Chen [76] discussed advances in the numerical modelling of CMEs. The CMEs in the simplest form may be considered as a three-part structure: a high-density core, surrounded by a dark low-density core, which is enclosed by the loop [77,78]. This three-part structure exists at a small scale in the active region, as well as at a large scale within a helmet streamer. The mass of the system is in the range of 1015–1016 g [79]. CMEs gradually lift out of the corona with speeds less than 100 km s$^{-1}$, or they may be impulsively ejected with speeds approaching 3000 km s$^{-1}$ [80]. The kinetic energy evolved is of the order of 1033 ergs.

The association of CMEs and ICMEs becomes more evident through STEREO obser-vations [27,81]. For example, the range of physical processes governing the evolution of a CME/ICME pair could be identified with the observed event during 12–18 December 2008 [62,81–83]. STEREO data show a connection between the solar eruption and the disturbance observed at the Earth. Figure 3 describes the STEREO observations of the 12 December 2008 CME/ICME event [84]. The observed CME was induced by a promi-nence eruption in the northern hemisphere (left panel of Figure 3) between 03 and 04 UT on 12 December 2008. The prominence material (visible in EUVI at 304 Å) was well aligned with the CME core. The CME slowly rotated and expanded toward the ecliptic plane, which is seen as fully developed in COR2. The basic structure of the CME remained organized (field of view of HI1). In HI2 of STEREO A, a dark cavity bracketed by structures with enhanced densities is seen. Figure 4 shows the plasma and magnetic signatures of the corresponding ICME, which passed wind on 17 December 2008. The shaded region identifies the magnetic ejecta of the CME [84]. Figure 5 shows the source region and evolution of CMEs observed on 16 June 2010 [85].

(2) ICMEs and Magnetic Clouds

The magnetic flux ropes are considered as a key feature associated with CMEs because there is (i) a correlation between ICMEs and magnetic clouds, (ii) a link between erupting prominences possessing a twisted magnetic field and CMEs, (iii) the appearance of helical structures in chronographs. The magnetic flux rope structure reproduces the three-part density structure of CMEs and many of its observed properties [30,86–88].

Flux ropes ejected from the solar corona associated with CMEs usually remain intact through interplanetary space, and careful examination suggests that they may be connected to the magnetic structures observed at 1 AU [81,83,89–91]. The magnetic fields associated with ICMEs may also retain flux rope structures, which are referred to as magnetic clouds (MCs) [92–94]. Usually, ICMEs are characterized by low plasma beta (<0.1), low ion temperature and high magnetic field strength, with a smooth rotation of the field direction. The rotation of the field favours the flux rope geometry [94–96] and the occasional presence of counter-streaming electrons support to the fact that the magnetic field remains attached to the Sun at both ends [97].

The relative proportions of ICMEs appear as MCs show large variations throughout the solar cycle. At solar minimum, nearly all ICMEs at the Earth can be identified as MCs [59,98], which reduces to ~15% at solar maximum, with an average of 30% over the solar cycle [99]. The cycle dependence reflects their place of origin, orientation and mutual interaction of CMEs. At solar minimum, usually, the majority of CMEs originate from

streamer blowouts and quiescent filament eruptions at low latitude, which may produce slow CMEs. In contrast, at solar maximum, more CMEs originate from active regions at high latitude. The eruptions are in off-centre ICMEs, which are less likely to register the field line rotations of a flux rope [88].

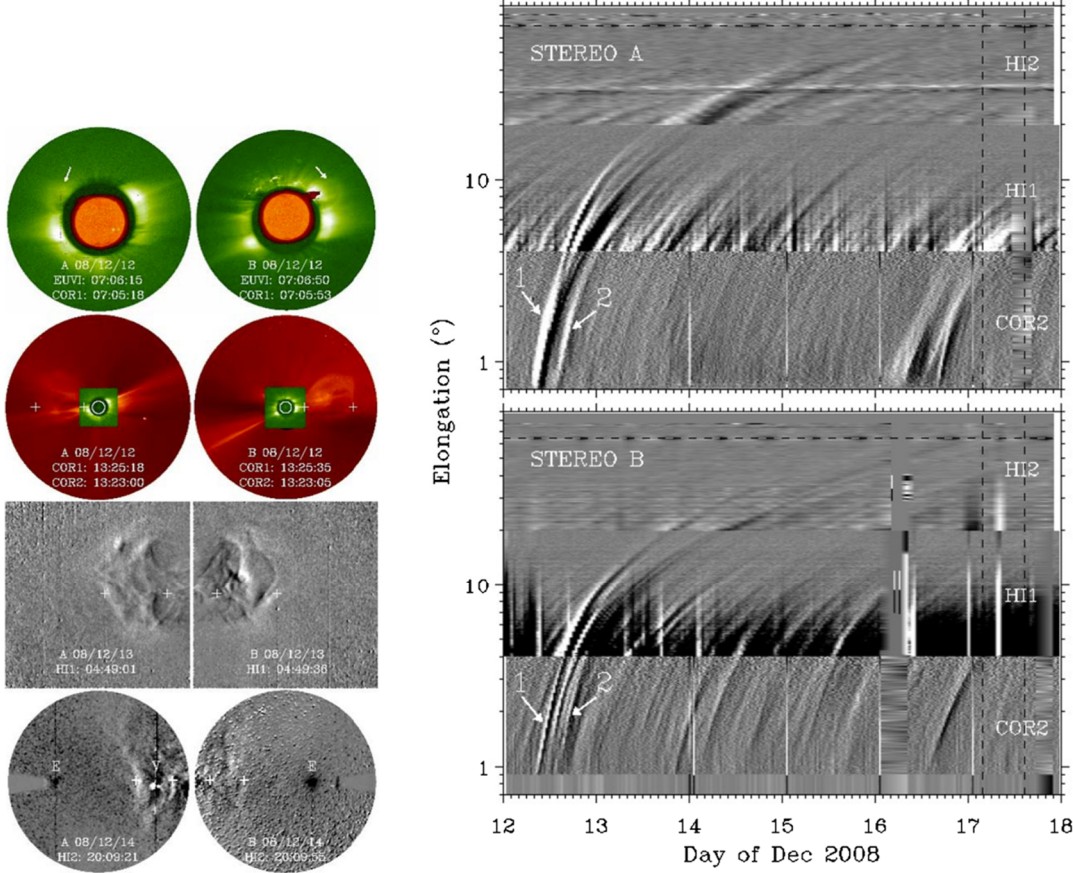

**Figure 3.** STEREO observations of 12 December 2008 CME/ICME event. The left column shows the CME/ICME evolution observed by STEREO A (**left**) and STEREO B (**right**). From top to bottom, the panels display the composite images of EUVI at 304 Å and COR1, showing the nascent CME (indicated by the arrow), combined COR1 and COR2 images of the fully developed CME, and running difference images from HI1 and HI2 when the ICME is far away from the Sun. The crosses mark the locations of the CME leading and trailing edges obtained from the time–elongation map. The positions of the Earth and Venus are labelled as E and V. Right: time–elongation maps constructed from running difference images of COR2, HI1 and HI2 along the ecliptic plane for STEREO A (**upper**) and B (**lower**). The arrows indicate two tracks associated with the CME. The vertical dashed lines show the MC interval observed at wind, and the horizontal dashed line marks the elongation angle of the Earth [84].

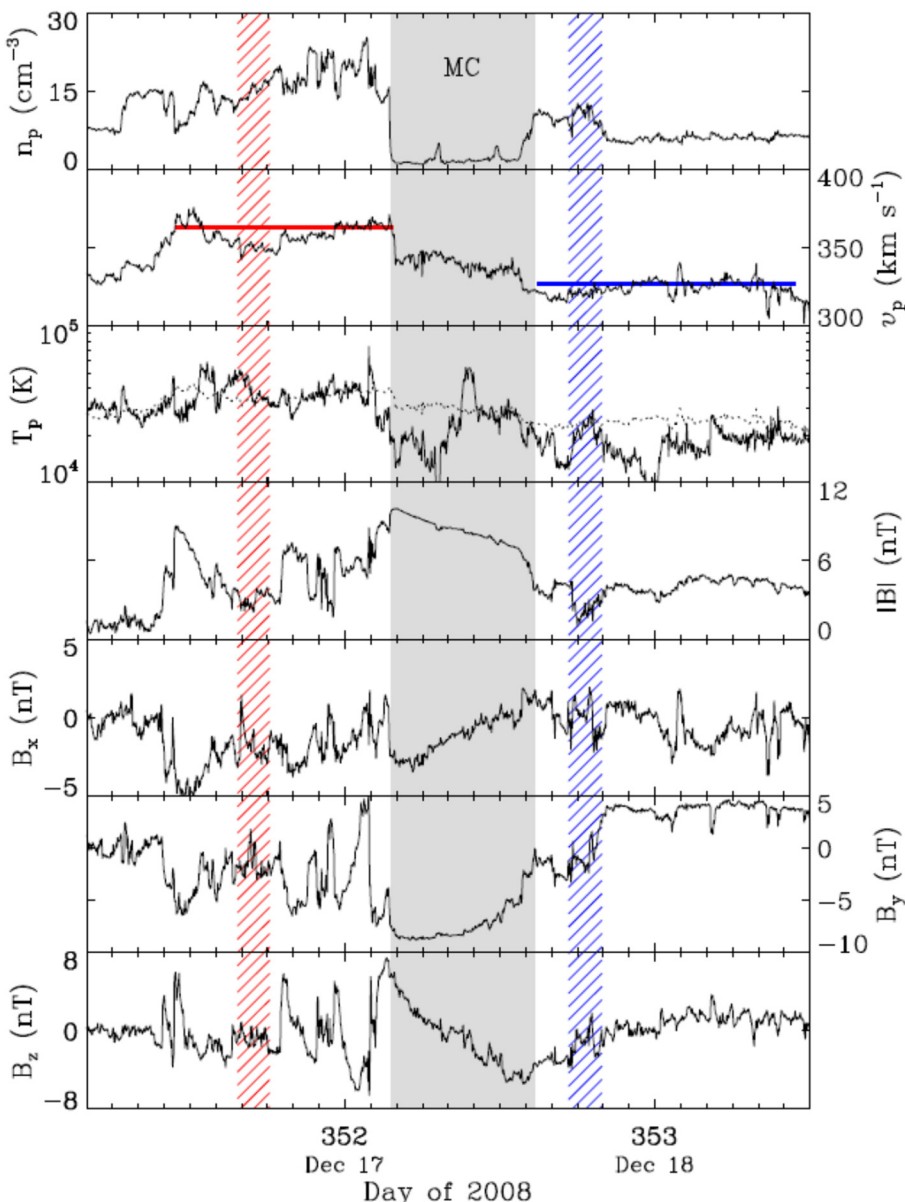

**Figure 4.** The MC observed at wind corresponding to the 12 December 2008 CME [84]. From top to bottom, the panels show the proton density, bulk speed, proton temperature, and magnetic field strength and components, respectively. The shaded region indicates the MC interval, and the hatched area shows the predicted arrival times (with uncertainties) of the ICME leading and trailing edges. The horizontal lines mark the corresponding predicted velocities at 1 AU. The dotted line denotes the expected proton temperature from the observed speed.

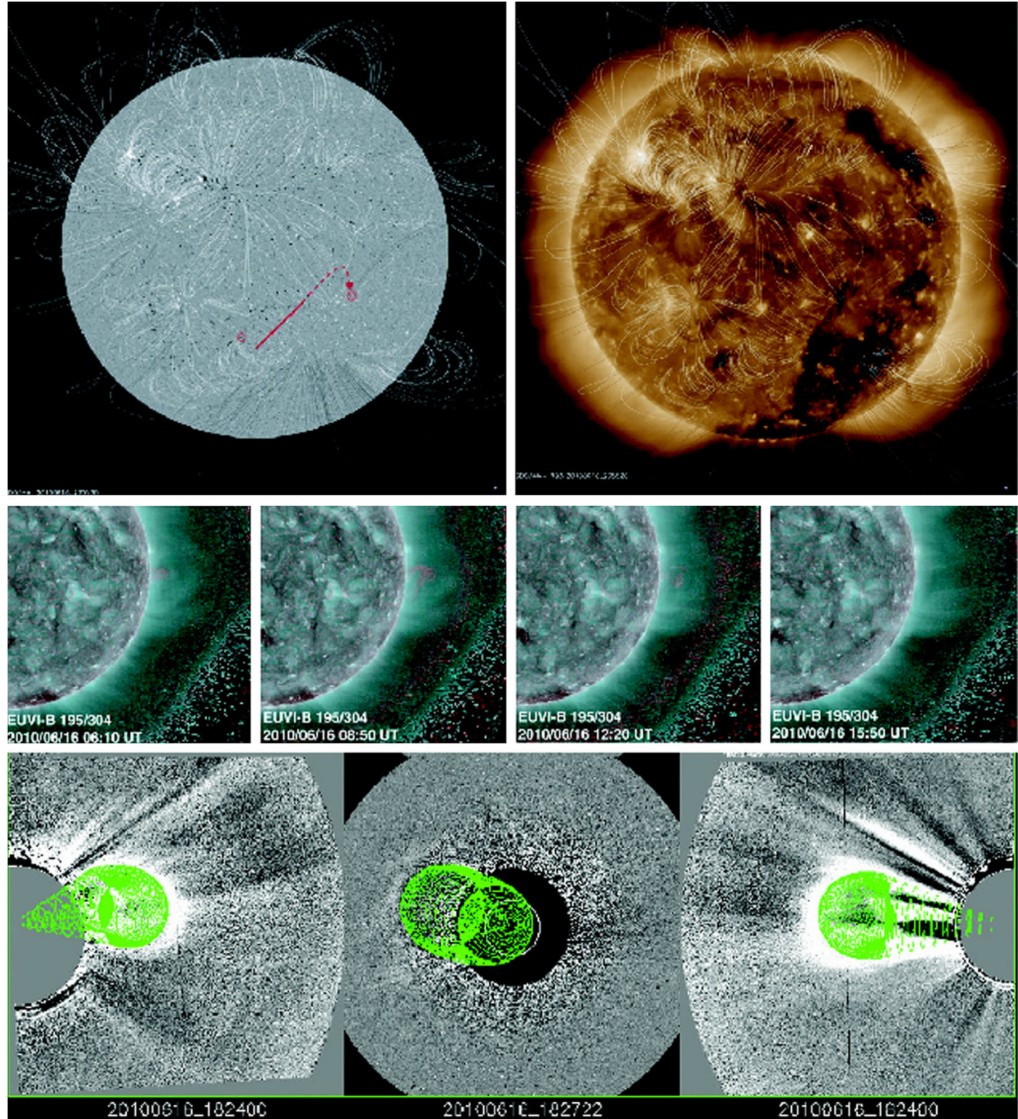

**Figure 5.** Solar observations of the source region and CME of 16 June 2010. Top left: the HMI synoptic magnetic field is shown with PFSS coronal extrapolation. The red line indicates the source of the CME red circles giving the foot points of the erupting flux rope. Top right: AIA 193 Å image of Sun. Middle row: STEREO EUVI-B images in 195 and 304 Å show a quiescent filament lifting off as the CME erupts. Bottom row: GCS model fitting to the CME event of 16 June 2010. The left, centre and right panels are simultaneous data from STEREO COR2-B, SOHO LASCO C2, and STEREO COR2-A, respectively. The images have been over plotted (green) with the GCS model represented by a grid of points on the surface of the model flux rope [85].

### 2.1.3. Solar Flares

Flares happen in the corona due to the nonpotentiality of the coronal field. White-light flares are responsible for flare energy release when high-energy particles from the corona precipitate to chromospheric/photospheric layers. The energy contained is in the form of radiative energy, kinetic bulk energy, thermal and non-thermal energy [100–102]. The spatial size of a flare varies from event to event, and the small event may have a flaring loop height of less than 104 km, whereas it may reach 105 km in the largest event [103]. The duration of a flare (103–104 s) and the amount of energy released during a flare depends on the size of the flare. Initial observations of flares were in white light [104,105]; however, they have been observed in a wide wavelength ranging from radio to gamma rays. Even the photosphere responds to a big flare, observed as a white light brightening [106,107].

Flares gain energy from the magnetic field during their movement through the convective zone of the Sun. The magnetic field embedded with plasma in convective motion forms twisted thin flux tubes [107–109]. Such a flux tube, after emergence from the surface, finds itself in a sudden decreased gas pressure and as a result, it expands rapidly. During this expansion, some magnetic energy is lost, and the magnetic structure in the solar atmosphere is formed. Part of the magnetic energy remains in the system as a field aligned electric current [110–112]. Observations indicate the presence of precursors in the form of an emerging bipolar region preceding the onset of a flare, which may interact with the pre-existing magnetic field in the corona and produce flares [113–117]. Another form of precursor is the activation, or eruption of a filament composed of a relatively cool plasma temperature ($\sim$104 K) floating in the corona containing a relatively high temperature ($\geq$106 K) plasma. The activation of filament destabilizes the magnetic structure containing filament. Since filament is formed in a low-beta plasma region (magnetic pressure is dominant compared to gas pressure), the main forces causing such destabilization are the magnetic pressure, gradient force, magnetic tension force, and gravitational force. These forces control the eruption condition of a filament, and hence, detailed study is required to understand the phenomena.

### 2.1.4. Solar Energetic Particles (SEPs)

The enhancements of particles (electrons, protons and heavy ions) in space with energies much larger than the average energy in the corona (~few hundred eV) are considered as solar energetic particles (SEPs). SEP events have a well-defined onset time and intensity enhancements up to several orders of magnitude. SEP events can last for several days with energies of few MeV/nuc and impulsive events last for a short duration ($\leq$1 day) with low intensity. In the period of high solar activity, impulsive events are numerous (~1000/year), whereas gradual events are ~few tens/year [118]. The SEP events emerging from western locations on the solar disk have much a higher probability to impact the Earth due to the curvature of the Parker spiral interplanetary magnetic field. Further, particles associated with SEP events, in some cases, can be detected at locations widely separated in longitudes [119,120] as well as in latitudes [121] during both cases of impulsive and gradual events [122]. Usually, SEPs from western events display a fast rise time with a shorter duration, whereas those from eastern events show a gradual rise phase, [123] while in some cases, the intensities of SEPs show a large peak at ~100 MeV during the CME-driven interplanetary shock that passes the Earth. This is termed as an energetic storm particles (ESPs) event and is observed in association with a weak soft X-ray (SXR) burst [124].

Further, regarding the association between SEP events and solar activity, observations suggest that all SEP events may not be associated with confined and eruptive solar activity [125,126] but few of them may be associated with CMEs [127,128]. Sophisticated statistical methods suggest that SEP peak intensities are correlated significantly with both the CME speed and flare parameters, especially soft X-ray fluence [118,129]. The forecast of SEP events are possible with long and short lead times. The long lead time (~days) forecast relies on the analysis of solar active regions and their magnetic configuration. The short lead time forecasts are based on the detection of solar eruptive events, including solar flare and/or a CME. Forecasting is important for the space weather evolution/mitigation and the assessment of its impact. There are two groups of models; namely, the empirical models based on known statistical relationships between various types and magnitudes of SEPs [130] and the physics-based models, which include relevant acceleration and transport processes [126,131].

### 2.1.5. Filaments (Prominences)

Filaments are formed in magnetic loops between two strips of opposite polarity that contain relatively cool and dense gas suspended above the surface of the Sun [132]. Filaments (prominence systems) appear in a variety of sizes and shapes, with their appearance depending on the magnetic environment. The filaments over a narrow polarity inversion

zone are usually very thin with no lateral structure and heights $\leq 103$ kms. The filament–prominence systems over a very weak magnetic network are usually much higher and more irregular in shape, with an intricate fine structure. In fact, all filament–prominence configurations share basic patterns in their magnetic structures. Observations of eruptive prominences are found to be consistent with an erupting flux rope model in the active region [133–135]. Depending upon the location of eruptions, prominences are referred as active regions (ARs). Those found in the active region are intermediate prominences, and those found adjacent to the active region or between active regions are quiescent filaments. Engvold et al. [136] classified these events based on morphology, dynamics, magnetic configuration and spectroscopic properties.

## 3. Space Weather: Terrestrial Perspective

### 3.1. Sun–Earth Interaction and Magnetosphere

The Sun affects the Earth's space environment through various plasma chains in the solar wind, electromagnetic radiation and high-energy charged particles accelerated in the solar corona. The electromagnetic solar radiations, apart from transferring information of solar activity and eruptions, ionize the upper layers of the atmosphere, forming the ionosphere. Strong X-ray flares may also produce observable ionization in the atmosphere [137] and may cause significant irregularities by introducing noise in the wireless communication systems [138,139].

ICMEs with the associated shocks and sheaths, fast solar wind, and fast/slow stream interaction regions (SIRs) are the main drivers of intense magnetospheric activities. Magnetic reconnection [140] is the key factor in energy, momentum, and plasma transfer from solar wind to the magnetosphere, and the southward IMF (BS) is a major contributor to the geomagnetic storm strength. In addition to refined measurements of particles and fields, numerical simulations are also being used to understand the dynamics of the Sun–Earth interaction. There are two approaches: one is the hybrid approach, where electrons are treated as fluid and ions are treated as quasi particles to apply in the particle-in-cell model [141]. The other method uses the distribution function as in the Vlasov hybrid model [142].

Van Allen Probe (VAP) observations have added a new dimension to radiation belt dynamics and to understanding the magnetosphere [143]. The energization through interplanetary shock-induced electric field impulses and wave particle interactions has been successfully applied to Van Allen Probe observations of various storm events [144]. During the storm period, the plasmapause is pushed closer to the Earth, resulting in plasma outflow, and ultimately, the ionosphere is enhanced and contributes to the ring current processes. The role of substorms is also important for space weather prospectives. Actually, the substorm process intensifies the magnetosphere–ionosphere interaction, as the magnetospheric plasma sources and produces seed population for high-energy electrons through wave particle interactions [145]. Space weather that affects human activity and harms technological infrastructures [139,146] is distinct from solar terrestrial and heliospheric impacts [36,147,148]. Severe mangnetospheric conditions are responsible for the direct risk of ground-induced currents (GICs) where current systems can cause rapidly varying magnetic field signatures at the Earth's surface [147,149–151].

### 3.1.1. The Solar Wind–Magnetosphere Interaction

The solar wind interplanetary magnetic field (IMF) controls the dynamics of the Earth's magnetosphere and is responsible for reconnections at the dayside magnetopause [140]. This further stresses the open magnetic field lines on the night side that leads to the formation of an elongated region called the magnetotail (Figure 6). Closed magnetic flux of the plasmasheet is transported back again towards the dayside, completing the magnetospheric convection cycle [152]. The ionospheric polar cap represents the region of magnetic field in the magnetosphere that is 'open' to the Earth at only one end. Dayside reconnection increases the size of the polar cap, as it reconnects solar wind plasma and

magnetospheric field lines to form open field lines, while the night side reconnection of open field lines in the northern and southern lobes reduces the size of the polar cap [153]. The rate of dayside and night side reconnections is not in balance [154]. This further means that the amount of open flux in the magnetosphere varies, as indicated by the orange field lines in Figure 6 [155]. This clearly indicates that energy stored in the magnetotail is subsequently released during the reconnections. Consequently, if the rate of reconnection at the dayside magnetotail does not match, the polar cap will change in size, and can expand or contract. The reality is more complex because of some modifying phenomena in the process. The polar cap grows in size before a substorm onset because energy is deposited into the lobes by magnetopause reconnection, while it reduces in size as a substorm occurs during night side reconnection. This overall picture is known as the expanding/contracting polar cap paradigm [55,155]. In brief, it may be interpreted as the extension of the open magnetosphere model to time-dependent behaviour where the reconnection rates in the magnetotail and on the dayside do not match [36].

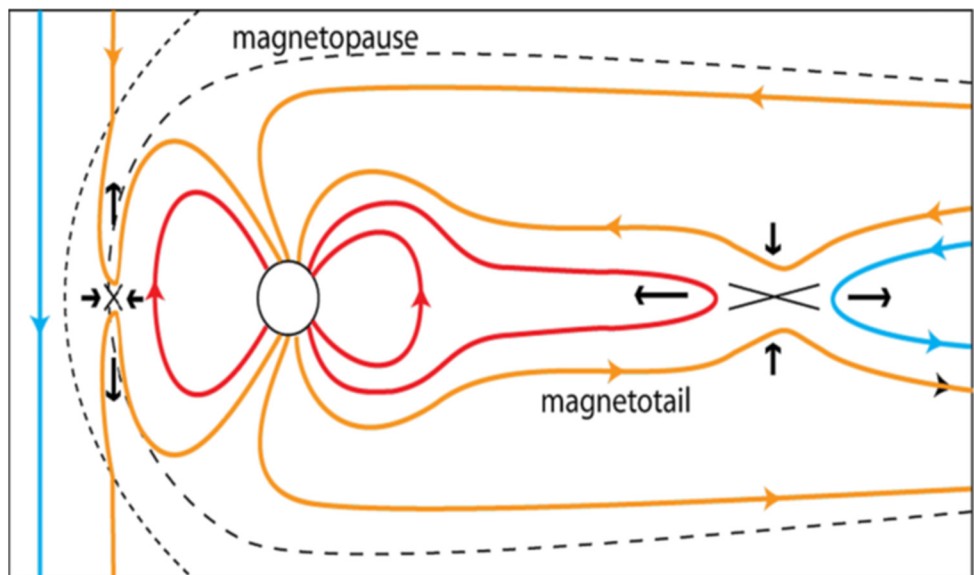

**Figure 6.** A schematic of plasma circulation in the Earth's magnetosphere for southward IMF conditions [36].

The internal processes in the magnetotail further leads to the formation of thin current sheets near the Earth; consequently, reconnections take place more strongly and result in drastic changes in the magnetotail configuration [156]. Figure 7 depicts various features of the processes involved in configuration. This further represents a cut through the magnetotail while the x-line and plasmoid does extend out of the page (limited in size). The dipolar region is confined to the central region of the magnetotail, which results in magnetosphere–ionosphere coupling [157].

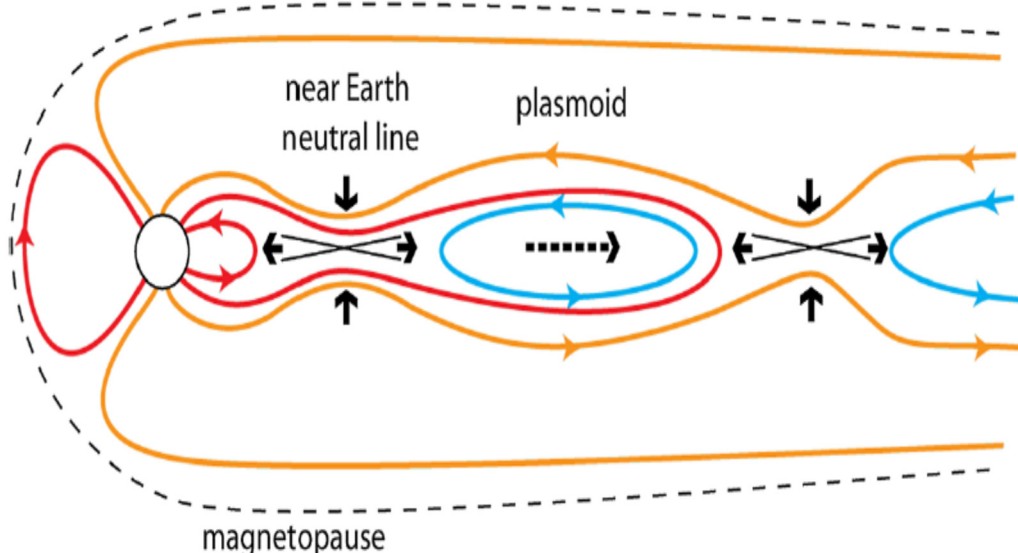

**Figure 7.** Formation of the near-Earth neutral line. This leads to the development of a plasmoid which is ejected down tail [36].

### 3.1.2. Magnetospheric Dynamics: Current Understanding

(1) Solar Wind Coupling at Magnetopause

The magnetosphere plays a decisive role in space weather variability. Solar wind and the magnetosphere coupling provide inputs for magnetospheric changes. In fact, the L1 Sun–Earth Lagrange point is the best location that measures inflowing solar wind related to magnetospheric space weather, and algorithms have already been developed for the same [38]. However, the variability in the solar wind perpendicular to the flow direction makes the situation very difficult [158]. Nevertheless, recent observations show that in the operational modelling of the propagation delays from L1 to the Earth, there is no significant difference between perpendicular and tilted solar wind phase fronts [159]. Conditions observed in the solar wind are not necessarily representative of those encountered upstream of the magnetopause and sometimes may lead to an incorrect prediction of the impact on the magnetosphere, where the IMF orientation significantly modifies the magnetosheath [160,161]. It has been shown that the magnetic field Bz component can have opposite signs upstream and downstream of the bow shock [162,163]. This is exemplified in Figure 8, which shows the Bz along the magnetopause for an IMF with a northward Bz, obtained from a magnetosheath model [164]. As depicted by the blue areas, Bz turns south in some parts of the magnetosheath, and reconnection occurs. The modification of the solar wind properties across the bow shock may also explain the saturation of the polar cap potential during strong magnetospheric driving. The polar cap potential is proportional to the solar wind electric field up to a certain limit, where it saturates and no longer responds linearly [55]. Pronounced dawn–dusk asymmetries in the magnetosheath parameters, such as the magnetic field strength and plasma velocity, the ion density and temperature, have been studied, and these asymmetries are associated with the Parker spiral IMF [165]. The draping of the field lines along the magnetosphere also contributes to the spatial modification of the magnetic field direction inside the magnetosheath. The spatial variation of the magnetosheath properties can affect the onset of magnetic reconnection and the development of the Kelvin–Helmholtz instability [166]. Additionally, the bow shock and magnetosheath properties can be significantly modified during low Alfvén Mach number conditions, which are often associated with ICMEs and magnetic clouds. This may alter the solar wind–magnetosphere coupling during extreme space weather events [163,167]. The fundamental challenge for magnetospheric space weather forecasting is to understand when and where reconnections will occur and the physics behind the occurrence of reconnections [168–170].

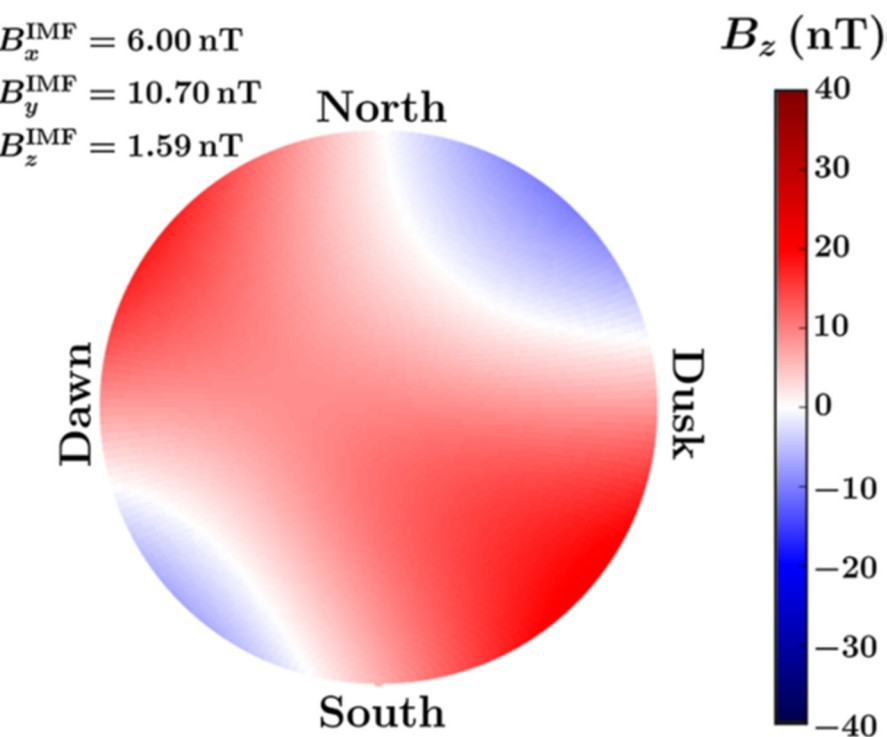

**Figure 8.** Magnetic field Bz component along the magnetopause for a weakly northward IMF semi-analytical magnetosheath model based on ideal MHD [164].

(2) Physics of Reconnection and Diffusion Region

Magnetic reconnection has large-scale magnetospheric consequences. The reconnection itself occurs within the small-scale diffusion region where the plasma decouples from the magnetic field and the frozen-in field. The diffusion region is structured because of their larger mass and volume over which the ions are not frozen in [171]. The physics of the diffusion region has been the subject of much recent investigation to understand precisely how the plasma and the magnetic field decouple. The formation of Hall magnetic fields due to the different motion of the ions and electrons is a key signature of the ion diffusion region which has received considerable attention [168,172,173].

Observations from the Magnetospheric MultiScale (MMS) mission (identical spacecraft flying in a tetrahedral formation) have focussed on the electron diffusion region (EDR), where each spacecraft carries an identical payload of instruments to measure the electric and magnetic field, the plasma population, the composition and the energetic particles [174]. The MMS fast particle instrument (FPI) measures the electron–ion all sky distribution function at 30 ms and 150 ms resolution, respectively [175], and is able to probe the electron physics and the properties of the EDR in detail. The first major success of the MMS mission has been to record multiple encounters with the EDR and to measure the properties of the electrons involved.

(3) Temporal Variability of Reconnection and Transient Behaviour

The magnetic reconnection can be steady at the magnetopause, where temporal variations are often observed. The flux transfer events (FTEs) are bipolar signatures in the component of the magnetic field along the magnetopause normal [176]. Although there are various production methods, recently, the multiple x-line reconnection has received more attention because it was observed through spacecraft data [177]. Thus, the flux transfer events are important in the context of magnetospheric space weather because they contribute to the transfer of solar wind plasma and the energy into the magnetosphere. Magnetic topology plays a key role here; if they are connected to the magnetosphere at only one end, then they will indeed transport flux into the magnetotail. However, they

may be connected to the magnetosphere at one end, both ends, or neither end, and flux rope topology is crucial to investigate the complexity [178]. An alternative approach is to use multi-spacecraft observations of FTE geometry to constrain their shape, size and the formation mechanism [179].

(4) Other Plasma Transport Mechanisms: Kelvin–Helmholtz Instability

It is important to recognise some other mechanisms that contribute to the transport of plasma into the magnetosphere, and Kelvin–Helmholtz instability (KHI) is one of them. It arises in connection with the large velocity shear on the flank magnetopause and causes magnetopause surface waves to steepen into vortices and break [180]. It will occur most easily if the field is perpendicular to the flow, and is therefore important for northward IMF conditions, although it has been observed for southward IMF as well [177]. In the nonlinear stage, microscale instabilities or reconnection may occur, enabling plasma to be transported across the boundary and which is further resolved by MMS observations [181]. Ultimately, the KHI is important for space weather prospectives, because it allows solar wind plasma to enter on the flanks during northward IMF and mass transport across the magnetopause appears to be enhanced for northward IMF [182]. Better quantification of the plasma transport rate from new measurements and simulations will improve knowledge of cold dense plasmasheet formation and magnetospheric 'preconditioning' prior to intervals of southward IMF driving geomagnetic activity.

### 3.2. Storm–Substorm Association

Storms are the dominant cause of space weather consequences. Arising during Sun–Earth interactions, storms lead to enhanced magnetospheric circulation, the injection of energetic particles into the inner magnetosphere, and enhancements of both radiation belt fluxes and the ring current [183–186]. Coronal mass ejections (CMEs) and corotating interaction regions (CIRs) are the other drivers of geomagnetic storm activity [187–190].

The substorm can describe the elemental behaviour of the magnetosphere, which has both magnetospheric and auroral manifestations [191,192]. The basic nature of substorms can be understood in terms of the overall Dungey model, modified by the expanding/contracting polar cap model, and the near-Earth neutral line model [140]. Magnetospheric substorms are relevant for magnetospheric space weather because the reconfiguration of the magnetotail is associated with dynamic auroral emission which is directly associated with a reconfiguration of currents coupling the magnetosphere to the ionosphere, as well as the release of energy from the magnetotail [186,193–195]. Substorms therefore involve both transient/localized signatures and also large-scale changes in the tail current sheet configuration and energization of the plasma. The Time History of Events and Macroscale Interactions during Substorms (THEMIS) mission in particular was designed to study the chain of events that lead up to the triggering of a substorm [196,197].

There are several current systems that connect magnetosphere and ionosphere [45]. During substorms, the magnetic field in the magnetotail near the Earth becomes more dipolar. This reduces the cross-tail current in a region transverse to the Sun–Earth line associated with the dipolarization. To ensure current closure, a so-called substorm current wedge is formed, where field-aligned currents arise which connect the magnetotail with the ionosphere [186,195]. The strength of these current systems is often characterised using ground magnetic field measurements. The dynamics of these various current systems that connect ionosphere to the magnetosphere can now also be monitored by satellites and the associated deflection of the magnetic field [198,199].

### 3.3. Magnetotail Dynamics

Magnetic flux transport in the near-Earth tail region is dominated by high-speed plasma flows called Bursty Bulk Flows (BBFs), which are considered to be the outflows of the near-Earth reconnections. Associated electromagnetic field disturbances transport energy closer into the Earth's inner magnetosphere and are responsible for energetic particle injection and also for enhanced auroral precipitation. These energetic particle signatures

of intense substorms are identified as important space weather phenomena that lead to spacecraft anomalies such as surface charging and deep dielectric charging [200,201].

### 3.3.1. Bursty Bulk Flows, Dipolarization Fronts and Dipolarization Flux Bundles

As mentioned above, Bursty Bulk Flows (BBFs) are high speed flows in the plasmasheet with a low occurrence rate. BBFs contribute significantly to the total magnetic flux transport in the near-Earth magnetotail [202]. BBFs have 10 min timescales, with embedded velocity peaks of 1 min duration called flow bursts [203]. They are localized in the dawn–dusk direction with scales of about 1–5 RE. In the flow braking region, both the Earthward propagating dipolarization front and subsequent tail ward progressing dipolarization can be detected [204]. The dipolarization front (DF) is a thin (800–2000 km) boundary often observed on the leading edge of a BBF where the Bz component sharply increases [205] while the dipolarization flux bundle (DFB) corresponds to the entire Bz enhancement following the dipolarization front [206].

The combined flow bursts (FBs)/DF/DFB entity can be regarded as a region of enhanced electric field so that the different ions and electrons in the plasmasheet, depending on their pitch angle and energy, can enter into this region for a limited time in a localized region of space as the front propagates Earthward, and are accelerated accordingly [207,208]. Not all, but a large fraction of the fronts were shown to be tangential discontinuities, meaning that at least for those boundaries, the dipolarization front is simply a thin boundary dividing two different plasma regions without net bulk flow across the boundary [209]. Particles that are trapped in the collapsing field line of BBF/DFB can gain energy from Fermi/betatron acceleration, as was simulated and observed behind the dipolarization front [210]. On the other hand, other cases of dipolarization fronts were reported in which ambient ions pass the front layer, accelerated by the electric field at the dipolarization front and then reflected back ahead of the front [157,211]. While the occurrence of the rapid flux transport rate observations significantly drops between 10 to 15 RE [203], the dipolarization front grows (Bz amplitude increases) closer to the Earth, and a large dawn-to-dusk electric field in the flow braking region is both predicted from simulations and reported from observational studies [212]. Energetic particle injections accompanied by DFBs that were found from the midtail reconnection region (30RE) to inside geosynchronous orbit also suggest the importance of the transient/localized electric field enhancements of BBF and flow braking for the particle acceleration process [213,214].

The total energy gained from this process is, however, limited by the strength of the dawn-to-dusk-induced electric field and the width of the dipolarization front. For a typically localized dipolarization front case, the above mechanisms explain acceleration of a few keV to hundreds of keV for electrons, and a few tens of keV to hundreds of keV for protons [215]. These particles are considered as the seed population of the radiation belt particles that are subsequently further accelerated by different mechanisms such as by waves in the inner magnetosphere. Observation of a global injection front accompanied by MeV particles detected inside 4 RE during storm time conditions indicates that the maximum energy may be enhanced during those active events [216]. Such observations suggest that for an intense substorm case, DF/DFB/BBF-related acceleration may also contribute significantly to particle acceleration in the inner magnetosphere/radiation belt.

### 3.3.2. BBF Size and Magnetotail Flux Transport

From the considerations of global flux balance, it is required that BBF should have an effective dawn–dusk size of about 10 RE [217]. Large-scale size in the azimuthal direction is also expected from the size of the substorm current wedge. Furthermore, a typical timescale for the substorm expansion phase is about 30 min, which is again longer than that of typical DFBs. One explanation is that large-scale substorm effects are the integrated effects of multiple BBFs that take place in space and time. Multiple flow bursts and associated auroral precipitation have been considered as supporting evidence for this [218]. Temporal and spatial relationships between these BBF-associated field-aligned currents and the

substorm current wedge are further discussed by Kepko et al. [219]. Although only for a limited number of cases, multipoint studies have shown evidence for a dipolarization front extended to 10 RE scale during an intense substorm in the flow braking region and during storm time intense substorm in the inner magnetosphere [216]. The longer timescale of the substorm current wedge or auroral electrojet as compared to BBFs/DFBs, on the other hand, could also be due to the accumulated effects of the flow bursts that modify the pressure distribution in the inner magnetosphere and thereby sustain the substorm current wedge [219].

### 3.3.3. The Effects of Preconditioning on Magnetotail Dynamics

Magnetotail disturbances are modified by both the preconditioning of the magnetotail and the solar wind driver. The background magnetic field configuration, plasma density and temperatures, are important factors for determining the magnetotail disturbance as well as the magnetosphere–ionosphere coupling. Sergeev et al. [220] have showed that the solar wind electric field is associated with a larger auroral electrojet when the plasmasheet is hotter and has lower density than for the colder and denser plasmasheet. This difference could be due to the stronger field-aligned acceleration of precipitating electrons for higher temperature and lower density, as expected from Knight's relationship [221]. The hot, low density plasmasheet is expected to be produced by the lobe plasma entering and heating by reconnection. On the other hand, the source of the cold and dense plasmasheet is thought to be the solar wind/magnetosheath plasma entering from the flank [180]. Although minor in the number density, the effect of ionospheric heavy ion outflow can further contribute to changes in the timescale and changes in the spatial scale of the current sheet due to larger inertial scales [222].

The general effect of larger dawn-to-dusk solar wind electric field, created by southward IMF, is to move the near-Earth reconnection region closer to the Earth [51]. However, the response of the magnetotail to southward IMF conditions also depends on the nature of the preconditioning and the mode of magnetosphere–ionosphere coupling can differ from the simple substorm cycle, such as during steady magnetospheric convection (SMC). During the steady magnetospheric convection, enhanced and stable convection persists longer than a typical recovery phase with no substorm expansions [223]. The average total pressure in the inner magnetosphere is higher during the steady magnetospheric convection events than for other types of activity [224]. This higher-pressure region extends to larger radial distances, and causes fast Earthward flows to divert toward the dawn or the dusk flanks and continue to the dayside, leaving the inner magnetosphere relatively quiet. This pattern of flux transport is in contrast to substorms, during which, flows are directed toward the inner magnetosphere. The most extreme case of southward IMF Bz is the storm time substorm, when significant loading of the magnetotail leads to the reconnection site moving closer to the Earth [216,225,226].

### 3.4. Ring Current and Space Weather

Ring current is a toroidal-shaped westward flowing current around the Earth at geocentric distances between 2 and 9 RE with varying density. It consists of electrons and ions (mainly H+, O+, He+) with energies from about 1 KeV to 400 KeV (222). The current intensity during quiet time reaches ~1–5 $nAm^{-2}$, which may increase up to ~50 $nAm^{-2}$ during storm time [227,228]. The radiation belt storm probes (RBSP) mission produced extensive observations of ring current [43,229–232]. The source of ring current particles is inward moving plasmasheet particles under the action of time-varying electromagnetic fields. Plasmasheet itself is populated by particles from the terrestrial ionosphere and the solar wind. There are open questions about this mechanism, especially during storm and substorm time. On the other hand, there is argument that large-scale convection can alone account for the build-up of ring current [233].

During storm time, the ring current distribution shows asymmetric behaviour. This arises due to the asymmetric compression of the magnetosphere by the solar wind dynamic

pressure, which on the dayside compresses the magnetosphere while stretching the same on the night side. This results in highly asymmetric plasma pressure distribution during disturbed conditions due to asymmetric plasma transport and nighttime injection from the night side plasmasheet into the ring current [234,235]. Figure 9 presents the evolution of storm time ring current in the equatorial plane for the storm of 22 July 2009 computed with the Tsyzanenko TO2 global magnetospheric magnetic field model [236].

**Figure 9.** Evolution of the inner magnetosphere current density in the equatorial plane computed from the curl of the magnetic field modelled with the Tsyganenko T02 global magnetospheric magnetic field model during the storm event on 22 July 2009. (**a**) A relatively symmetric ring (**b**) the ring current intensifies with the development of the storm. (**c**) the asymmetric ring current, with its peak shifted duskward. (**d**) During the recovery phase of the storm the ring current density decreases, (**e**) but remains asymmetric. When Dst decreases intensified ring current is seen. (**f**) with Dst = −57 nT, the ring current density decreases, and distribution shifts toward symmetry, although it is not fully symmetric.

A relatively symmetric ring current at distances inside 5 RE along with the near-Earth part of the tail current at distances of 10–15 RE is seen (Figure 9a) just before the onset of the storm (Dst = 2 nT at 00 UT). At 04 UT, Dst = −27 nT, the ring current intensifies with the development of the storm (Figure 9b). When the storm reaches its peak at 06 UT, Dst = −80 nT, the ring current becomes asymmetric, with its peak shifted duskward (Figure 9c). During the recovery phase of the storm at 08 UT (Dst = −69 nT), the ring current density decreases (Figure 9d), but remains asymmetric. When Dst decreases further to −75 nT at 0915 UT, again, intensified ring current is seen (Figure 10e). At 1800 UT, with Dst = −57 nT, the ring current density decreases, and distribution shifts toward symmetry (Figure 9f), although it is not fully symmetric. Though the ring current density decreases, it is still higher than the non-disturbed level [236].

### 3.4.1. Ring Current Electrons and Effects on Satellites

Moderate energy (<100 KeV) electrons are important constituents of the ring current. The assessment of their contribution during storm time varies from 10% based on the

radiation belt storm probe (RBSP) measurements [237] to 19% based on Explorer 45 electron data [238]. Using OGO 3 satellite data, Frank [239] reported about 25% of ring current energy during storm time. These electrons may not penetrate the satellite surface but could pose serious risk though surface charging [240]. The interaction of ring current-charged particles and solar UV radiation with the surface material leads to the generation of secondary electrons. In order to have a zero net current between the surface and the incident plasma, the surface material of the satellite will be charged and surface will have some voltage. Usually, the sunlit area of satellite surface will be positively charged, and shadowed area negatively charged.

Electrons have higher speeds, so they are the main source of initial plasma current to a satellite. However, photo and secondary electrons depend on the surface material, and it may be higher than the plasma electron current to it. There could be discharge between different adjacent materials because of different potential levels or between the surface material and space of structured ground. The electrostatic discharge can electromagnetically couple into electronic circuits and may cause damage. Observations suggest that most satellite anomalies due to surface charging at geostationary orbit occur during night and early dawn [241,242]. This may be due to hot plasma injections during substorms from the magnetotail into the night side inner regions. The injected plasma may undergo gradient and curvature drifts towards dawn and may cause significant changes in the satellite charging levels.

### 3.4.2. Ring Current and Inner Magnetosphere Populations

Ring Current and Radiation Belts

The ring current often overlaps with outer radiation belt plasma particles and plasma systems have distinct populations with very different dynamics and responses to solar wind forcing [243–245]. The ring current causes the loss of radiation belt particles during the main and recovery phases of geomagnetic storms due to pitch angle scattering [246] and the adiabatic Dst effect [247]. The electromagnetic ion-cyclotron (EMIC) waves driven by unstable ring current proton distributions [248] efficiently interact with high-energy radiation belt electrons [246] and cause their precipitation.

Simulations using the Van Allen Probes data [247] demonstrated the dramatic effect of EMIC waves on >1 MeV electron pitch angle distributions [248]. The simulation results show that the flux levels of electron populations are reduced substantially at lower pitch angles, resulting in significant precipitation. Electron precipitation may impact on human activities in the biosphere as well as to terrestrial weather and climate systems [249]. In the outer radiation belt, the interaction of whistler mode (chorus) waves with low energy anisotropic electrons (~10 keV) and the wave grows at the cost of temperature anisotropy distribution [250]. Chorus waves may interact with ring current energetic electrons (~10–100 keV) through cyclotron resonance and may accelerate electrons to very high energy [251,252].

Ring Current and Plasmasphere

The ring current overlaps plasmasphere with its peak coinciding with the plasmapause boundary [253,254], and thus, variation in ring current can affect the shape of the plasmasphere, including the location of the plasmapause boundary [255,256]. Thus, enhanced convection during a geomagnetic storm may result in the direct transport of plasmaspheric particles to the outer magnetosphere, resulting in the erosion of the plasmasphere on a very fast timescale [257,258]. The variations in plasmaspheric dynamics during contraction and refilling periods can trigger lower hybrid wave growth [259,260] and whistler mode wave growth, which may scatter electrons and ions closer to the Earth. Ring current injections influence the storm time, and thus, affects the ionospheric convection pattern, which may have space weather effects on human technologies [261–263].

### 3.4.3. Ring Current, Ionosphere and Below

The ring current during storm time influences the ionospheric flow and perturbs the density configuration at mid-latitudes [264]. During storm time, anomalously high ionospheric electron densities on the dayside are reported at mid—high latitudes, which influences the global positioning system (GPS) and other communications [265–267]. Total electron content (TEC) maps derived from GPS signals have been extensively used to quantify the timing and intensity of storm enhance density (SED) [268], which have been found to extend through the dayside auroral zone into the polar cap [269,270].

During storm time, inner magnetosphric plasma becomes highly asymmetric and localised field aligned currents (FACs) are generated due to the pressure gradient, which create sub auroral polarization streams (SAPs) that affect the neutral upper atmosphere. Wang et al. [271], based on a statistical study of the influence of SAPs on the thermosphere, reported about a 10% increase in density within the flow channel. Wang et al. [272], based on numerical modelling, found that SAPs cause increases in the global thermospheric temperature, but this effect takes days to fully develop.

The heating caused compositional changes in the thermosphere, with the upwelling of the molecular-rich atmosphere and downwelling of atomic oxygen-rich gas elsewhere. The localized thermospheric density and temperature enhancement may create atmospheric gravity waves [273]. Field-aligned currents lead to the coupling of large-scale ionospheric electric currents to the magnetosphere. At lower latitudes, FACs are mapped to the ring current region [274,275]. Thus, the storm time geomagnetic-induced current (GIC) is re-enforced. Usually, GICs with large amplitudes are mainly considered at high latitudes, but recent past observations [276–278] show enhanced ring currents during storm time at mid and low latitudes.

### 3.5. Space Weather Effects on the Earth

### 3.5.1. Extreme Space Weather

Severe space weather events are usually associated with very complex solar activity, such as solar radio bursts (SRBs), white light flares, X-ray emissions, solar proton emissions, fast solar wind, shocks, CMEs and ICMEs. These events have severe consequences on the biosphere, ground/satellite communication system, satellite navigation, spaceship, etc. The enhanced X-ray causes additional ionisation in the ionosphere, which may perturb signal propagation through the ionosphere, whereas SRB may introduce noise in the receiver [279,280]. An intense space weather event may have wide societal consequences, such as the case of March 13, 1989, when the peak Dst was about −640 nT [281], which caused the collapse of the Hydro-Québec power distribution system. In another event of 23 July 2012, the CME did not hit the Earth. The transit time to 1 AU was about the same as in the Carrington event, and the peak IMF as observed by STEREO was larger than 100 nT. According to existing models [282], the expected Dst could have been ~−1200 nT. However, the validity of the model to so large an event has not been examined. Further, according to the standard magnetopause models based on pressure balance, this event may have pushed the dayside magnetopause almost down to the ionosphere [283,284]. Even in this prediction, the counter action of the induced current and saturation of polar cap potential are not considered. However, the event could have wiped a significant fraction of plasma and energetic particles away from the magnetosphere. These examples just show the severity and magnitude of destruction that can occur on the planet during extreme space weather events, although their occurrence frequency is very small.

### 3.5.2. Effect on Technologies

Technological systems based on the Earth's surface and in the near-distant places in the space are dramatically affected by space weather events. The magnitude of the effect depends on the intensity of the event and the way it interacts with the systems in the Earth's environment. A fundamental phenomenon behind the most important ground-based effects is electromagnetic induction leading to an induced current in the system called

geomagnetically-induced currents (GICs). GICs may cause damages to power systems, pipelines and telecommunication systems [285–287].

The GIC may disturb power systems through the saturation of transformers [288], an increase in the exciting current and harmonics in the electricity, unwanted relay trippings, excessive reactive power demands, a blackout of the whole system, and maybe, the permanent damage of transformers. The damage level of the induced current in transformers depends on the structure of transformers and on other engineering components in the particular power network. The storm-induced current varies from system to system and, in a system, from site to site. For example, the largest measured GIC was 600 A in Sweden during the geomagnetic storm in April 2000 [289], but the correct value seems to be 'only' about 320 A [150]. Another example is the complete damage of the Hydro-Quèbec high-voltage system in Canada during a geomagnetic storm [290]. In the same storm, a transformer was damaged due to overheating and had to be replaced in New Jersey, USA. However, the magnitude of the GIC is not available. In another storm in March 1991, the GIC reached 220 A at a substation in northern Quèbec [149] which caused some problems; however, it was insignificant compared with the March 1989 case. This suggests that magnitude of the GIC during the storm of March 1989 may have been enormously large.

The geomagnetic storm may also lead to the current flow from unprotected buried pipelines into the soil [291], and thus, the electrochemical corrosion of pipes may occur. This may be avoided by protecting steel pipe from soil contacts either through insulating coatings of steel pipe or through a cathodic protection system, which keeps the pipeline in a negative voltage in the order of 1 V with respect to the ground. Telecommunication systems with metallic wire connection could also suffer from GIC harm [292], including fires [293]. The use of optical fibre cables may reduce the possibility of damage. However, in addition to optical fibres, the power for a repeat station uses metallic wire, which remains sensitive to storm-induced current. Wallerius [294] reported the malfunctioning of traffic lights on Swedish railways during a geomagnetic storm in July 1982.

### 3.5.3. Space Weather—Societal Impacts

Space weather has two major elements relevant to society: one is scientific research, and the other is applications. The high-energy particles from the Sun during space weather events adversely affect the operation of the spacecraft flying around the Earth by degrading the solar power arrays, damaging the sensitive electronics and even the payloads, etc. The excess radiation in the space can be harmful to the astronaut also. However, for people on the Earth, the health effects will be very limited depending on their location on the Earth and the severity of the event [295].

In addition to the damage of spacecraft, power transmission, pipelines, traffic signals, human health, etc., the drastic change in ionospheric electron density can disturb the HF radio communications satellite-based ground navigation. This adverse effect may be seen at the high-latitude aviation, where the aircraft crossing the polar regions may have to divert their path to the lower latitudes. The enhanced electron density in the signal path may lead to a loss of signal power. If the reduction in power goes below the threshold level of the ground receiver, then the receiver may not be able to lock these signals properly, and hence, disruption in the estimation of user position and navigation path is caused. In addition to this, the enhanced EUV radiations during flare events may heat the neutral atmosphere causing its expansion, and hence, introduces the extra drag on the near-Earth orbiting satellite, which reduces the life span of the low Earth orbiting satellite [34].

### 3.6. Anthropogenic Space Weather

In addition to effects caused by the variation in the solar activity, even humanmade activities may also lead to some changes in the atmosphere/ionosphere, including the ground effects. During the Cold War period (1958–1962), the U.S. and the U.S.S.R. ran high-altitude tests with exotic code names such as Starfish, Argus and Teak. These nuclear explosions affected the electromagnetic environment of the Earth to a large extent [296].

### 3.6.1. Geophysical/Geomagnetic Signatures of High-Altitude Nuclear Explosions

The detonation of explosives during the Cold War period at heights from 26 to 400 kms above the surface mimicked some of the natural effects discussed earlier. Detonation caused an expanding fireball of hot plasma, which caused distortion in geomagnetic field lines and induced an electric field on the surface [296]. The electrons generated in the fission process may be trapped in the Earth's atmosphere/ionosphere by the geomagnetic field and could enhance and change plasma properties, and this may cause all the relevant effects produced by the solar events. Herbert York conducted experiments in 1958 and showed that the injection of electrons could damage spacecraft flying through such a cloud of anthropogenic space weather [297]. The nuclear explosion may also produce electromagnetic pulses (EMPs) which could be very dangerous. Such a devastating EMP was observed after the Starfish Prime event of 9 July 1962 [298]. High-altitude nuclear explosions could produce a variety of EMPs, which may have widely different effects depending upon the geomagnetic location and burst altitude [299]. The gamma rays produced during nuclear detonation interact with the surrounding medium and generate Compton electrons. The gamma rays moving downwards face exponentially increasing air density, and thus, knockout electrons with increasing density. The Earth's magnetic field causes the electrons to turn coherently, looping around the magnetic field, and thus, the current may generate an EM signal [300]. The work process is shown in Figure 10 [301].

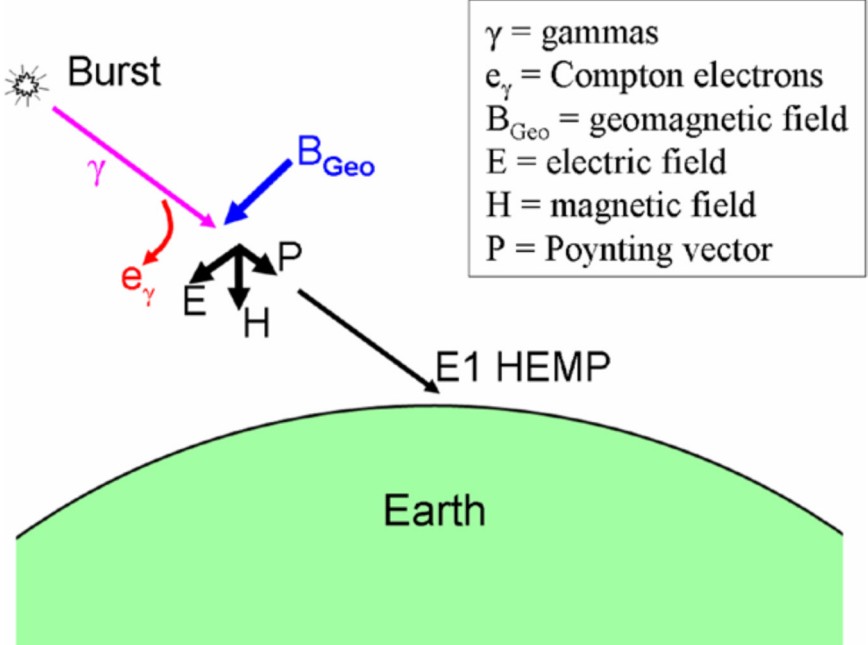

**Figure 10.** General basis of the EMP generation process. Gammas from the nuclear burst interact with the upper atmosphere generating Compton electrons, which are turned in the Earth's geomagnetic field, and produce a transverse current that radiates an EM pulse towards the Earth [301].

Geomagnetic disturbances caused by the Starfish event were recorded in the United States [302,303], Canada [304], Australia [305], India [298] and many other places. After about 40 years, Dyal [306] made a comprehensive analysis of data from sounding rockets carried out at the time of Starfish events (near the Johnston Island site) and reported the dynamics and evaluation of the diamagnetic cavity formed immediately after the explosion. The cavity, formed in 1.2 sec after explosion, reached a length of 1840 km along the magnetic field line and 680 km in height, reaching beyond 1000 km above the explosion site. The bubble collapsed in about 16 sec [306]. The results reported by Dyal [306] suggest that the observation of a nuclear explosion may yield new results if properly reanalysed in the presence of the present understanding of the subject.

### 3.6.2. Space Weather Effects on Anthropogenic VLF Transmissions

VLF radio waves can penetrate seawater a few hundred feet, so have been proved to become a powerful tool as VLF transmitters for submarine communications using very high-power beam radiated from antenna complexes [307]. Clilverd et al. [308] presented a list of this variety of active VLF transmitters at distributed locations with power ranging from 25 kW to 1 MW. A portion of the transmitter energy escapes into space, and governed by ionospheric conditions, it may contribute to scatter radiation belt electrons. These waves were further used for probing the ionosphere and the magnetosphere [139,309]. These waves can also trigger a variety of natural emissions while interacting with the existing particle population in the region. Ionospheric heating and perturbations in electron/ion densities associated with VLF-transmitted waves have also been observed [310]. The transmitted wave may reach beyond the plasmapause under certain conditions and may amplify, showing a dramatic increase in wave intensity [308]. This was revealed by the Van Allen Probes wave data [311] and the event was termed as a VLF bubble. The outward extent of VLF bubbles may correspond to the inner edge of the Van Allen belt.

The quenching of natural VLF hiss emission in the magnetoshere by monochromatic carrier waves of high-power VLF transmissions [312] and the amplification of VLF transmitted pulses in the magnetosphere storm periods [313] are also observed. In addition, VLF transmitters produce perturbations in electron/ion densities and the heating effect in the ionosphere, and thus, can contribute to space weather.

### 3.6.3. Artificial Radiation Belts

The discovery of inner and outer radiation belts, usually called Van Allen belts, was the first major scientific discovery of the space age [314–316]. An inner radiation belt contains electrons in the energy range 100 keV–1 MeV, while an outer radiation belt includes electrons up to much higher energies of multi MeV. Energetic charged particles may cause operational problems in spacecraft, usually called single-event upsets (SEUs). SEU effects can be caused by galactic cosmic rays, solar energetic particles, or trapped ions in the Earth's inner Van Allen radiation belt. Moderate-energy electrons may produce surface differential charging effects, and high-energy radiation belt electrons could induce deep-dielectric charging conditions [317]. Surface charging is usually associated with about 10–100 keV electrons during geomagnetically active times when spacecraft surfaces are in shadowed regions.

Christofilos [297] proposed the modification/creation of an artificial radiation belt and its impact on satellites and other space applications. Hess [318] reported damages to six satellites that sustained damage from the energetic particles generated by Starfish. Wenaas [319] and Conrad et al. [320] presented more extensive lists of damages. The radiation data from the Telstar sensors along with the electron radiation effect data from the laboratory measurement could explain the malfunction and failure of the satellite's command system [321]. The investigations that explored surface damaging are not clear about the radiation damage process [319]. Modern spacecraft electronics have become very highly precise, and the components used are more sophisticated and prone to damage, even for small deviations from the normal conditions, and hence, the impact of a similar severity may cause incalculable damage to the infrastructure, including space technology/human society.

### 3.7. *Solar Energy Variations and Climatic Change*

The Sun is the principal source of energy in the solar system, and hence, the climate/weather in the biosphere is governed by the energy input from the Sun. As described earlier, space weather is the conditions in the space which is governed by the processes occurring on the solar surface [4,112]. Therefore, even a small change in any of the solar transients is responsible for variable weather conditions in the Sun–Earth system, which may affect the climate conditions of the Earth [112].

Research outcomes have revealed the existence of two areas in which space weather might influence the global climate change [322]. The first is the solar origin, which relates the small variations in total solar radiation [323], and the second is the investigation of impacts by kilometre-size bodies on the Earth's surface, which are entirely unrelated to the solar flux variations [205,324]. Solar energetic particles (SEPs) are the other factor responsible for space weather conditions that can impact the Earth's atmosphere. These particles penetrate into the atmosphere and change its chemical constituents, such as ozone (O3), nitrous oxide (NO), etc. These changes in minor species can have long-lasting consequences in the middle and upper atmosphere, and ultimately have a significant impact on the global climate of the Earth [325,326]. Sometimes, the solar minimum conditions also affect the Earth's climate because during the solar minimum, the maximum amount of high energetic particles reach the Earth's atmosphere and create nucleation sites in the atmosphere which seeds cloud formation and create cloudier conditions [327–330].

### 3.7.1. Solar Irradiance Measurements

The total electromagnetic radiation output from the Sun, called the solar constant, has been monitored continuously by full spectrum cavity radiometers on board seven spacecraft for the last 20 years [324]. The average value of the solar constant at the mean distance of the Earth from the Sun is 1370 Wm$^{-2}$. This value has shown an intrinsic solar variation over the course of various solar cycles of ~0.1%, but it is uncertain how indicative these solar radiations affect solar variations. Figure 11 has depicted the variations of the total solar irradiance during the last forty years. There have been many works that attempted to identify the source of variations [331–334]. Mostly, the solar irradiance variations originate in solar surface magnetic activity and approximately 80% of the measured variation has been accounted for sunspots and faculae [335]. Various mechanisms have been adapted to link solar activity with the dynamics of the terrestrial climate and related calculations [322,336].

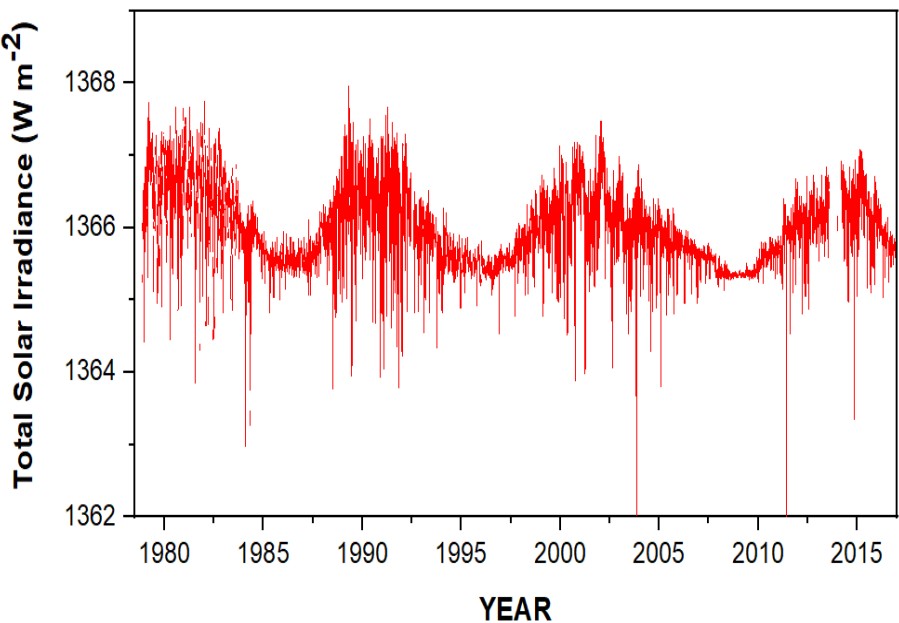

**Figure 11.** Solar irradiance variations as recorded by a composite of satellite sensors for the period 1978 to 2018. (Available online: lasp.colorado.edu/) (accessed on 10 March 2020).

### 3.7.2. Solar UV Irradiance

Solar Ultraviolet (UV) radiation also plays its role in space weather activity, and ultimately, on climate change conditions [337,338]. The direct effect of irradiance variations is amplified by an important feedback mechanism involving ozone production, which is

an additional source of heating [337,339]. The origin of the lower stratospheric maximum and the observed signal that penetrates deep into the troposphere at mid-latitudes are less well understood and require feedback/transfer mechanisms both within the stratosphere and between the stratosphere and underlying troposphere [340].

### 3.7.3. SEPs and Galactic Cosmic Rays

The other mechanisms involved are solar energetic particles (SEPs) events and galactic cosmic rays (GCRs). SEPs are generated at the shock fronts ahead of the major solar magnetic eruptions and penetrate the Earth's magnetic field over the poles, where they enter into the middle atmosphere (thermosphere, mesosphere) and on rare occasions into the stratosphere. A large fraction of SEP ions are protons (sometimes called solar protons), but they are accompanied by a wide spectrum of heavier ions [341]. SEPs cause ionization, dissociation, and the production of odd hydrogen and nitrogen species that can catalytically destroy ozone [52,342]. Recently, Bhargawa et al. [343] studied the impact of solar energetic protons over the total ozone column during super storms (Dst index < −300 nT) and have quantitatively estimated the variations of total ozone column and UV radiations by applying superposed epoch analysis (SEA).

The solar activity can modulate the cosmic rays and ultimately the cloud formation (weather), and hence, might be a viable factor for the Sun–climate mechanisms [112,327,344,345]. Since GCR flux is enhanced during solar minimum, it increases atmospheric ion production, and thus, the ion-induced formation of sulphate aerosol can act as an efficient cloud condensation nucleus (CCN) as a possible route by which the atmospheric ion changes could influence cloudiness. Further, GCR–cloud links have been proposed through the global atmospheric electric circuit [330,346,347]. The global circuit causes a vertical current density in fair weather, flowing between the ionosphere and the surface. This fair-weather current density passes through stratiform clouds, causing local droplet and aerosol charging at their upper and lower boundaries. Aerosol charging modifies the cloud microphysics, and hence, as the current density is modulated by cosmic ray ion production, the global circuit provides a possible link between solar variability and clouds [348,349].

## 4. Space Weather Predictions and Future Prospective

### 4.1. Space Climate—Long-Term Variations

Space climate is controlled by the Sun, which is variable in space and time, and hence, causes variation in the space weather phenomena. Changing patterns have been reported in solar cycle length, maximum and minimum activities in the form of sunspot numbers [16,350]. Even sophisticated statistical and physics-based models are unable to exactly forecast the cycle and its various features. For example, the peak of solar cycle 23 was not as strong as most of advanced models had suggested, and also the long (2008–2010) and low valley before solar cycle 24 was not predicted at all. Based on Ulysses observations, McComas et al. [50] reported that during the minimum before solar cycle 24 the fast solar wind was significantly less dense, cooler and carried less mass and momentum flux compared to the previous minimum, which was not predicted by models. The tendency of reduction in the open magnetic flux observed in cycle 23 continued even through cycle 24.

The sunspot numbers show that solar cycle 24 seems to be weakest cycle since solar cycle 14. The question is whether the solar dynamo is undergoing some kind of long-term change resembling the Maunder minimum [351] or if this a signature of the Gleissberg [352] cycle, or if it is just a question of 'statistical fluctuations'. One of the consequences of decreasing solar activity is enhancement in the galactic cosmic ray fluxes in the inner heliosphere, which may lead to enhanced levels of radiation doses, resulting in hazardous risks both in space and at airline altitudes. This is, evidently, a serious concern for space activities and especially to missions for other planets.

### 4.2. Space Weather Predictions: Current Status

Based on the data analysis of several dedicated satellite missions such as GOES, NOAA POES, SOHO and ACE at (L1 point), SDO and the STEREO are issued at regular intervals during 24 h and 365 days. However, these alerts can be provided only after the solar material traverses about 98% of the distance in ~10–15 min in advance during the severe space weather events. However, since the Sun–Earth environment is so complex, our understanding is still only partial. An international programme CAWSES (Climate and Weather of the Sun–Earth System) sponsored by SCOSTEP (Scientific Committee on Solar–Terrestrial Physics) was in operation during 2006–2013 with an aim to enhance our understanding of the Sun–Earth environment and its impacts on society and life. The Indian counterpart of CAWSES, sponsored by ISRO, named CAWSES-INDIA, with an aim to bring the various Indian institutes and universities under one umbrella and promote space weather research in the country, was made operational. As the CAWSES had come to an end during 2013–2014, the subsequent programme, Variability of the Sun and Its Terrestrial Impact (VarSITI), sponsored by SCOSTEP (2014–2018), focused on the relevant peculiar solar activity and its consequences on the Earth. Presently, the PREdictability of the Solar–Terrestrial Coupling (SCOSTEP-PRESTO) programme is underway for 2020–2024. PRESTO's goals are to address the predictability of space weather events on two folds. The first is the prediction of events at the Sun, in the heliosphere and in the Earth's magnetosphere, ionosphere and atmosphere at timescales from seconds to days and months. The other one is the prediction of sub-seasonal to decadal and centennial variability in the Sun–Earth system.

### 4.2.1. Future Developments: Modelling and Forecasting

Various models have been developed in order to understand different aspects of solar wind–magnetosphere/ionosphere/atmosphere interactions [353–356]. Many of the developed models are part of the space weather modelling framework (SWMF), which brings together simulation models dealing with the whole chair of physical processes relevant to the space weather between the Sun and the Earth [357,358]. NOAA's Space Weather Prediction Centre uses the SWMF model operationally. However, the complexity of involved multidimensional phenomena needs the availability of multiple magnetospheric models at the operational level to provide a diversity of forecast methodology. Knipp [359] developed ensemble forecasting to speed up the predictions. Another option for improving forecasts could be to perform data assimilation. Data assimilation has actually been performed for the radiation belts and electron ring current using VERB-3D and 4D and Van Allen Probes data, which require large and accurate data at fixed intervals in space/time. Further, intense storms cause the movement of the dayside magnetopause towards the Earth; therefore, changes in the treatment of the inner boundary need to be incorporated. Even the development of appropriate metrics to test models in a relevant way becomes a challenge.

In the standard approach of coupling different codes describing relevant regions such as the ring current, the plasmasphere and the ionosphere/thermosphere systems, forecast accuracy has been improved; however, more improvement is required. One route could be to include reconnection physics by modifying the treatment of resistivity, or to embed particle in cell (PIC) simulation capability within the larger fluid simulation technique [360,361]. A global Vlasov code, Vlasiator, has been developed [142,362,363], which could be the ultimate choice as a simulation tool because of its sufficiently large computing power and accessibility. This technique may be useful in exploring the Axford conjecture, which argues that large-scale magnetospheric dynamics are governed by global processes and boundary conditions rather than the precise nature of the local resistivity, etc., associated with reconnection [364]. Currently, we are more dependent on space-based technologies, which are vulnerable to conditions in space weather, and hence, present/future generations would be more susceptible to space weather conditions. As a result, space weather specifications and forecasting requirements would become an integral part of our life.

In improving space weather forecasting, remote sensing technology is a twenty-first century revolution, where the continuous and global sensing of the neutral Earth's atmosphere is widely used in Numerical Weather Prediction (NWP). With more and more dependence on space technology, a similar revolution is expected in the modelling of space weather. At present, the accurate specification and forecasting of space weather phenomena is very difficult because (1) an accurate model dealing with coupling between the Sun, the magnetosphere, the ionosphere, the thermosphere, and the mesosphere is not available, (2) continuous, reliable and accurate observations of all of these regions are absent and (3) the ability to assimilate the data into the models in an optimal and self-consistent manner is not developed. These are great challenges for the space weather community and require tremendous efforts to fill in the gap.

### 4.2.2. Space Weather Prediction by Cosmic Rays

Cosmic rays (CRs) are highly energetic charged particles from solar and galactic origins, traversing the interplanetary medium to the Earth's environment. During this travel path, their interaction with intervening medium from the source to the destination (Earth) is seen through large effects of the solar activity and solar wind disturbances. GOES satellite observations provide some information about solar cosmic rays' properties and their impacts on the magnetosphere. The ground-based neutron monitors from worldwide networks provide information about galactic cosmic rays (GCRs). By improving instrumentations, the quality of data is significantly improved [365]. Presently, real-time data collection using the latest networking techniques are used to achieve reliable maximum data through the best synchronization and expandability [366]. The new IP-based network systems are very useful in joining all neutron monitor stations spread over worldwide and quality data collection in real time, which could be very useful for real-time data processing and forecasting.

Cosmic rays travel much faster than solar wind, and hence, the effect of solar wind disturbances on cosmic rays may reach well in advance of their arrival at the Earth. Using proper analysis/measuring techniques, the signature of solar wind in cosmic rays may be selected and the same may be used as an input to space weather applications [367]. Real-time data in combination with developed and tested methods may yield successful prediction with improved accuracy [368].

### Solar Proton Events

The Earthward-directed solar proton events usually escape from satellite detection with enough accuracy because of their small detection area cross section. However, ground-based neutron monitors can record with a statistical accuracy (in average, 0.5% for 5 min) as high as ground level enhancement. Hence, an alert of dangerous particle flux with the minimal probability of false alarm can be made using neutron monitor stations.

### Geomagnetic Storms

Real-time cosmic ray data may be useful in the study of solar transients heading towards the Earth and their probable impacts on the terrestrial environment. Cosmic ray density and anisotropy vary significantly during the arrival of solar disturbances [369]. For example, approaching coronal shock may accelerate particles and result an anomalous pitch angle distribution and also show specific features, such as: (a) a decrease in the CR intensity within a narrow range of pitch angles close to the interplanetary magnetic field direction and (b) a large, sometimes >1%, difference between the CR intensity from these and from other directions. Belov et al. [367] discussed that a pitch angle distribution which departs from the sum of the first two spherical harmonics may be used as an early indicator of an approaching disturbance and as a predictor of a geomagnetic storm. Leerungnavarat et al. [370], based on the analysis of ground-based cosmic ray data, showed that during a large heliospheric storm, significant variations in CR density and in the first harmonic of the CR anisotropy were observed, and simultaneously, a dramatic change in the interplanetary

and geomagnetic parameters occurred. The heliospheric storm was indicated by different space weather parameters.

### 4.2.3. Approaches to SEP Forecasting

Making forecasts with long lead times (of the order of days) relies on the precise analysis of the development of solar active regions and their magnetic configuration. On the other hand, short lead time forecasts are issued after the detection eruptive event, such as a solar flare/a CME. Presently, models are developed for the short lead time forecast, which aim at the prediction of SPEs from the observed eruption and its impact on the Earth's environment. Some earlier results have been summarized by Vainio et al. [131].

### Empirical Models

Empirical models are usually based on statistical relationships between various categories and magnitudes of solar events. For solar energetic protons (SEPs) prediction, NOAA Space Weather Prediction Centre (SWPC) developed the PROTONS system using an algorithm that takes the GOES X-ray peak flux of the flare into account and information on its location [371]. Kahler et al. [372] developed the Proton Prediction System (PPS), based on flare parameters. The COronal Mass Ejections and Solar Energetic Particles (COMESEP) is also being used as an alert system for forecasting the space weather impact based on empirical and statistical relationships [373], though it also uses a physics-based model to forecast event parameters [374]. The SEP prediction model of the University of Malaga (UMASEP model) uses the simultaneous tracking of profiles of soft X-rays and protons observed by the GOES spacecraft [375], and predicts the occurrence, peak time and peak intensity of an SEP event. This model has been incorporated within the SEPs FLAREs forecasting system [43]. The detection of relativistic electrons, or protons at 1 AU which travel faster than ions, have also been proposed for the forecast of protons [376].

Besides the above-mentioned models, a number of empirical relationships are available that may be exploited in future operational models. For example, the hard X-ray bursts observed by the Solar Maximum Mission (SMM) exhibit a distinct spectral hardening (i.e., flattening) during the SPE events [377], which was also confirmed by RHESSI observations [109]. This is a distinctive feature, because in typical hard X-ray bursts, the photon spectrum starts soft, hardens as the intensity rises, and softens again in the decay phase [378]. Another important property of SEP events is that at energies above 100 MeV, they were accompanied by microwave bursts with spectra that had peaks at or above 15 GHz [379], whereas on average, microwave peak frequencies are around 10 GHz. This may be attributed with the fact that SEP events are associated with particularly strong electron acceleration, leading to high densities of radio-emitting electrons in the active flare region. When examining the soft X-ray bursts associated with SEP events, it was noticed that in the relatively low temperature event in the 0.05–0.4 nm channel of the GOES monitoring instrument, compared to the 0.1–0.8 nm channel, are more strongly associated with SEPs than the others [380].

Apart from empirical models, physics-based models based on the relevant acceleration and transport processes are also available. The SOLar Particle Engineering Code (SOLPENCO) solves a one-dimensional transport equation coupled with MHD modelling of the shock accelerating the SEPs. The Energetic Particle Radiation Environment Module (EPREM) within the Earth Moon Mars Radiation Environment Module (EMMREM) framework couples a 1D transport equation with a convection-diffusion equation to describe transport in 3D [381]. The SPARX model, based on the test particle approach, solves SEP trajectories in three dimensions to forecast time profiles of particle intensities at 1 AU [374]. Numerical models are also attempted to couple realistic simulations of CME propagation in the corona with particle simulations to describe the associated particle acceleration [86,382].

## 5. Space-Based Observations

Space-based satellite observations have tremendously contributed to space weather activities and research. Space is more than a thousand times emptier in comparison to the best laboratory vacuums located on the Earth, and the Sun fills it with fields, plasma and a tenuous wash of particles (available online: science.nasa.gov/heliophysics/focus-areas/space-weather) (accessed on 10 March 2020). In the past, various space missions such as the Solar and Heliospheric Observatory (SOHO, 1995), the Advanced Composition Explorer (ACE) (1997), the CLUSTER, 2000, the Solar and Terrestrial Relations Observatory (STREO), (2006), the Solar Dynamics Observatory (SDO, 2010), etc., have already contributed immensely to exploring the space environment around the Earth and throughout the solar system.

Missions such as the Van Allen Probes (VAPs) [383], which observe the radiation belts around the Earth, the Deep Space Climate Observatory (DSCO) for solar wind [384], Magnetospheric MultiScale (MMS) Mission [174] to study the Earth's magnetosphere using four identical spacecraft flying in a tetrahedral formation, Parker Solar Probe (PSO) [385] with the objective of making observations of the outer corona of the Sun and the Solar Orbiter (SolO) [386] launched on 10 February 2020, are successfully operational to perform detailed measurements of the inner heliosphere and nascent solar wind and are particularly focused on improving our understanding about the space weather phenomena. These missions are devoted to observing the Sun and the space environment almost $24 \times 7$ and provide information for the study of the solar atmosphere/dynamics. Future missions, such as L5, which is the fifth Lagrange point of the Sun–Earth gravitational system, would experience a stable orbit and remain in the same place relative to the Earth and the Sun. In addition, research missions such as NOAA's Space Weather Prediction Centre and the UK Met Office provide support for the improvement of space weather prediction models.

Space weather studies on other planets also have assisted significant advancements in the recent past [387–391]. The observations of escape processes from Mars Express (2003), Messenger (2004), Venus Express (2005) and the missions such as Mars Atmosphere and Volatile Evolution (MAVEN, 2013) have provided interesting comparisons of the Earth with different values in the medium parameters and have promised great insights for future space research programmes.

## 6. Discussion

Solar eruptions travelling towards the Earth are the main sources of impact on the environment of the near-Earth space. Modern society is moving away from nature and become more dependent upon ground-based and space-borne technologies, which are sophisticated, precise, tenderer, and hence, are more vulnerable to space weather events. Therefore, to mitigate the impact of such events, long-term continuous observations, forecasting, nowcasting and further investigation are required. To that end, high-quality data, proper models with efficient computation and the precise understanding of involved physical processes are required.

In order to have high resolution real-time observations of the Sun–Earth system, global networks of both the ground-based and the space-borne experiments could be developed at a modest cost, allowing real-time communications within a benign environment. A wide range of ground-based instruments useful for space weather observations are magnetometers, ionosondes, riometers, phased array systems and cosmic ray monitoring systems. Ground-based monitoring arrays spread all over the world with a common code and similar specifications are a critical component of any space weather observing system and should complement space-based observations.

The Earth is enveloped by the magnetosphere, which acts as a protective shield, preventing harmful solar wind particles and radiation entering deep into the Earth's atmosphere. Therefore, the precise understanding of various phenomena taking place in this region and surrounding zone becomes essential for the mitigation of space weather conditions. The present understanding of the relevant solar phenomena driving space

weather and their consequences in the near-Earth space could be predicted to some extent, but not always. In fact, the involved phenomena and variability at different temporal scales is too complex to be resolved from the present knowledge and modelling capabilities. Hence, advanced observational data analysis and numerical modelling are to be the priority for future. Observations of space weather in space, whether they are through remote solar processes or the in situ measurement of charged particles and electromagnetic fields, are very challenging technologically and economically and require cooperation among leading institutions. Usually, observations from space are often difficult to repeat, which is generally considered as one of the backbones of the scientific method. The Van Allen Probes (VAP) observations have provided great illustrations of the need for fresh data with improved instrumentation.

The source of space weather events is far away, and it is not possible to directly probe the physical processes, at least not continuously, and hence, some proxies are needed, such as empirical estimates of energy input from the solar wind to the magnetosphere. This is based on known energy output mechanisms and theoretical ideas of the input processes. To obtain the proxies, statistical analyses are usually used, which in this field are based on few samples. In case of space weather events, this aspect becomes the most important from a practical point of view. The other aspect is the complicated interdependencies of observable phenomena, which may or may not have causal connections, for example, SPEs vs. CMEs, SPEs vs. solar flares, or CMEs vs. solar flares.

For numerical modelling and forecasting the space weather events, much more improvement is required for the Sun to the terrestrial atmosphere. Some of the challenges are really serious, because physical processes often have their roots in the spatial and temporal scales of electron–ion motions. There are real problems associated with the inclusion of ion kinetics effects in the fluid models. Additionally, the vast range of relevant physical parameters to be included in modelling are a serious issue, in particular from the solar surface to the corona, where much of space weather originates.

## 7. Summary

Today, we are at an exciting stage of the development and forecasting of space weather events and providing services to common people. A number of models are now in operational use. The SWPC predicts the arrival of solar wind disturbances, such as CMEs in the Earth's environment, using the WSA-Enlil solar wind model. This is the first phase of a long-term transition in which the accuracy and usefulness of space weather services may increase through improved physical models, and validation of the models against observations.

The other key priority for space weather research is to improve the modelling of solar wind–magnetosphere interactions and the widespread applications, including the threat to power grids and pipelines. Another priority could be the generation and propagation of high-energy particles, the killer electrons in the Earth's radiation belt and auroral electron precipitation and their inclusion in models of solar radiation storms, because they are a significant source of surface charging on spacecraft in low polar orbits.

In the longer term, researchers are keen to address areas where we currently have only a limited understanding of the science, but where major advances in physics-based modelling could have significant potential. One example is the precise understanding of the development of magnetic activity inside the Sun and its connections to CMEs and solar flares and how to detect the whole/relevant processes. There is already some progress at the individual level, but the challenges are to make the connections across the full chain of events. This integration almost certainly needs to be carried out statistically, reflecting the fact that solar eruptive activity exhibits self-organized criticality. Therefore, models are required that can provide probabilistic forecasts on the time and scale of eruptions; in particular, the likelihood that a fast CME will be launched in a direction that could bring it close to the Earth. Solar wind propagation models can then be used to explore whether that CME will actually reach the Earth.

The development of new technologies for the betterment of human society is sometimes capable of reducing space weather effects. The good news is, that the use of higher frequencies (>4 GHz) in satellite communications increases band width and reduces disruption during space weather events. At the same time, a number of emerging applications, such as the likely growth in systems that embedded the use of satellite navigation, such as driverless cars, road tolling, and the control of rail systems, may be at risk from space weather events, which may cause these systems to experience positional errors due to unexpected changes in ionospheric delay, and the loss of signal due to ionospheric scintillation. Therefore, the communities developing these applications should be made aware of the space weather effects so that these impacts can be mitigated by a mixture of good design, forecasting and nowcasting.

## 8. Some Unanswered Questions

Observations suggest that the solar corona is much hotter that the solar surface. However, despite much observational, theoretical and modelling progress, the mechanism involved remains a largely unsolved problem. The reason may be the complexity of processes in the solar atmosphere that hinders the identification of simple and unique heating mechanisms [392], although scientists began facing this problem from the early 20th century [393,394]. The existence of corpuscular flow from the Sun [395], recognised as magnetized plasma [396] was explained by Parker [397] using a physics-based model for plasma escape from the Sun. Space-based measurements confirmed the existence and basic properties of the plasma chain connecting the Sun and the Earth. However, the response of the magnetosphere, as observed during an intense geomagnetic storm, is very diverse, and it is not clear how solar wind can shake the magnetosphere so strongly. Further, during solar wind–magnetosphere interaction, it is unclear how solar wind conditions control the magnetosheath conditions and low-latitude boundary conditions [398]. The majority of energy transfer and field-aligned currents to the ionosphere occurred through the boundary conditions. A physics-based understanding of these coupling processes is required for space weather models.

SEPs are found to have energies from a few keV to times of GeV and different space weather aspects are associated with different energy ranges. In the entire energy spectrum, it is impossible to separate impulsive and gradual SEP events. SEPs are accelerated during its journey from the source region to the Earth's environment. The relative contribution of acceleration at CME shocks and in small-scale processes involving magnetic reconnection depend on the considered particle energy [399,400]. Presently, it is not clear at which range the two mechanisms operate. Dierckxsens et al. [373] suggested a transition range of 10–20 MeV, with a stronger dependence of SEP intensities on CME speed at energies below 10 MeV and the opposite at energies about 20 MeV. Further, the effects of particle acceleration and particle transport are mixed, and it is essential to disentangle them. This can be possible only when measurements closer to the Sun are carried out.

Nevertheless, there remain many unresolved questions about the solar dynamo(s). For example, our knowledge about the subphotospheric meridional circulation, a key ingredient for solar subphotospheric dynamos, is very little. There is limited knowledge about the magnetic field in the polar region of the Sun, which, in association with meridional circulation, plays a significantly critical role in resolving the solar dynamo action. There seems to be two separate dynamos operating in the Sun: a deep-seated helical dynamo, responsible for the generation of the strong magnetic fields of active regions, and a near-surface chaotic dynamo, producing weak network fields. The interaction between these two dynamos and their roles in the overall solar cycle is not known.

**Author Contributions:** Conceptualization, A.K.S., A.B., D.S. and R.P.S., writing—original draft preparation A.K.S., A.B., D.S. and R.P.S., writing—review and editing A.K.S., A.B., D.S. and R.P.S. All authors have read and agreed to the published version of the manuscript.

**Funding:** This research received no external funding.

**Acknowledgments:** The authors are thankful to various data providers. A.B. is thankful to the University Grants Commission (UGC), India for providing financial support, the Rajiv Gandhi National Fellowship (RGNF).

**Conflicts of Interest:** The authors declare no conflict of interest.

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
