# Peer review of "Physics of Space Weather Phenomena: A Review"

_geosciences, doi:10.3390/geosciences11070286_

Round 1

Reviewer 1 Report

1. Please refrain from using VAP for Van Allen Probes and use RBSP. 

2. Please correct the text where MMS is referred to as "recently launched" as it has been more than 6 years. 

3. Please correct how abbreviations are introduced. 

4. Traditionally. empirical models are not based on neutral networks, please consider rewriting this sentence. (Summary)

Author Response

Reply to Reviewer 1 comments:

We are thankful to the learned reviewer for his critical and beneficial comments. The whole manuscript is rewritten.

The detailed replies are provided below:

  1. Please refrain from using VAP for Van Allen Probes and use RBSP. 

Suggestions incorporated in revised manuscript.

  1. Please correct the text where MMS is referred to as "recently launched" as it has been more than 6 years. 

Corrections are made.

  1. Please correct how abbreviations are introduced. 

Corrected.

  1. empirical models are not based on neutral networks, please consider rewriting this sentence.

Sentence is rewritten.

  1. (Summary)

Summary is rewritten.

Reviewer 2 Report

The work here presents a useful overview of the plasma populations in the magnetosphere and solar terrestrial interactions. The introduction motivates the review in terms of conveying information of space weather effects to policy makers, but the review would also be useful for new space weather students or anyone entering the field. In this respect, I would suggest that if the target audience is policy makers then more emphasis is placed on the consequences of space weather. 

There are a number of grammar and spelling mistakes in this work and I suggest that the authors take some time to carefully read through and check the paper. 

The review starts with an introduction section. There are several parts in this section which read somewhat strangely. For example:

line 39 - "...the terrestrial atmosphere, ionosphere, magnetosphere and interplanetary space, which house almost the entire modern communication system, transport system..." - is it necessary to say "almost" here? 

line 53 - 56 - the over explanation of a cycle waning from maximum to minimum is unnecessary. 

line 101 - 102 - It is not clear what you mean here.

I suggest carefully reading through and editing this section.

The review then continues to a section of solar-focused space weather. There are some typos in this section and occasionally acronyms are redefined or defined after use. 

Section 3 then moves to terrestrial space weather. This presents a board coverage of topics and is well balanced, not spending too much time on any one aspect. However, I have the following comments:

line 422 - ionosphere does not really influence radiation belt acceleration, at least not directly, as is implied. This is discussed correctly in later sections, but should be corrected here. 

line 425 - the ionosphere is not the only plasma source - it is the source for the plasmasphere, but radiation belt and ring current particles largely originate from the plasma sheet - as is discussed later in the review. 

Line 430 - it is not only ionosphere current systems which can contribute to GICs.

Line 445 - discussion of Fig 6 - The orange lines to not explicitly show the time variation you claim. Please reword. 

lines 496 - 498 - this doesn't make sense.

Line 522 - "simulations of space craft data" - it is not clear what is meant by this. 

"plasma sheet" appears as "plasma sheat" as several places in the manuscript - please correct. 

In section 3.4.1 - you might want to mention the difference in surface charging inside and outside the plasmasphere. 

Line 808 - astronomer -> astronaut

Section 3.5.3 - You mention the atmospheric expansion and the drag on LEO satellites, but you might want to also add a few lines on the the effect on space debris and the consequences for tracking.

Line 874 - Detoration -> Detonation

Section 3.6.2 - please explain that the large scale VLF transmitters are used to communicate with submarines. Also please explain that only a portion of the transmitter energy escapes into space, governed by ionospheric conditions, and this is the component that can then scatter radiation belt electrons. 

Line 921 - please point out that the inner belt is 100 keV - 1 MeV electrons and and the outer belt includes electrons up to much higher energies of multi MeV. 

Section 3.6.3 - this is labelled as 'Artificial radiation belts' yet the first two paragraphs discuss the natural radiation belts. In the final paragraph, discussing the artificial radiation belt, it is not clear that the radiation belt created from nuclear detonations is being discussed. 

As with section 2, section 3 has some acronyms which are defined several times. I suggest carefully reading through the whole manuscript and checking for this. 

The next section discusses modelling and prediction.

Line 1075 - "All of the developed models are part of the space weather modelling framework..." - there are several models that are outside of this. For example GORGON MHD code or the British Antarctic Survey Radiation Belt code.

Line 1082 - 1084 - Data assimilation has actually been performed for the radiation belts and electron ring current using VERB-3D and 4D and Van Allen Probes data. See https://agupubs.onlinelibrary.wiley.com/doi/10.1029/2020JA028208 and https://agupubs.onlinelibrary.wiley.com/doi/full/10.1029/2018SW002110.

The paper then concludes with a discussion, summary, and some open questions.

Author Response

Reply to Reviewer Comments: 

We are thankful to the learned reviewer for his critical and beneficial comments. The detailed replies are provided below:

There are a number of grammar and spelling mistakes in this work and I suggest that the authors take some time to carefully read through and check the paper. 

 We have corrected each and every spelling mistakes and grammar. In fact whole manuscript is rewritten and English is much improved because it is corrected by an English literary person.

The review starts with an introduction section. There are several parts in this section which read somewhat strangely. For example:

line 39 - "...the terrestrial atmosphere, ionosphere, magnetosphere and interplanetary space, which house almost the entire modern communication system, transport system..." - is it necessary to say "almost" here? 

We have removed the word `almost`

line 53 - 56 - the over explanation of a cycle waning from maximum to minimum is unnecessary. 

We have written the sentence and unnecessary lines are removed.

line 101 - 102 - It is not clear what you mean here.

We have removed these lines.

I suggest carefully reading through and editing this section.

Done. 

The review then continues to a section of solar-focused space weather. There are some typos in this section and occasionally acronyms are redefined or defined after use. 

We have corrected these errors.

Section 3 then moves to terrestrial space weather. This presents a broad coverage of topics and is well balanced, not spending too much time on any one aspect. However, I have the following comments:

line 422 - ionosphere does not really influence radiation belt acceleration, at least not directly, as is implied. This is discussed correctly in later sections, but should be corrected here. 

We have incorporated the suggestion by the reviewer.

line 425 - the ionosphere is not the only plasma source - it is the source for the plasmasphere, but radiation belt and ring current particles largely originate from the plasma sheet - as is discussed later in the review. 

Sentence is rewritten.

Line 430 - it is not only ionosphere current systems which can contribute to GICs.

Corrections have been made.

Line 445 - discussion of Fig 6 - The orange lines to not explicitly show the time variation you claim. Please reword. 

The mentioned sentence is rewritten.

lines 496 - 498 - this doesn't make sense.

We have removed the lines.

Line 522 - "simulations of space craft data" - it is not clear what is meant by this. 

We have removed the word `simulations`

"plasma sheet" appears as "plasma sheat" as several places in the manuscript - please correct. 

Corrected.

In section 3.4.1 - you might want to mention the difference in surface charging inside and outside the plasmasphere. 

Done.

Line 808 - astronomer -> astronaut

Corrected

Section 3.5.3 - You mention the atmospheric expansion and the drag on LEO satellites, but you might want to also add a few lines on the the effect on space debris and the

consequences for tracking.

We have rewritten the paragraph.

Line 874 - Detoration -> Detonation

Corrected.

Section 3.6.2 - please explain that the large scale VLF transmitters are used to communicate with submarines. Also please explain that only a portion of the transmitter energy escapes into space, governed by ionospheric conditions, and this is the component that can then scatter radiation belt electrons. 

Paragraph is rewritten and suggestion incorporated.

Line 921 - please point out that the inner belt is 100 keV - 1 MeV electrons and and the outer belt includes electrons up to much higher energies of multi MeV. 

Corrections have been made.

Section 3.6.3 - this is labelled as 'Artificial radiation belts' yet the first two paragraphs discuss the natural radiation belts. In the final paragraph, discussing the artificial radiation belt, it is not clear that the radiation belt created from nuclear detonations is being discussed. 

Paragraph is rewritten.

As with section 2, section 3 has some acronyms which are defined several times. I suggest carefully reading through the whole manuscript and checking for this. 

 Corrections are made.

The next section discusses modelling and prediction.

Line 1075 - "All of the developed models are part of the space weather modelling framework..." - there are several models that are outside of this. For example GORGON MHD code or the British Antarctic Survey Radiation Belt code.

Correction incorporated in revised manuscript.

Line 1082 - 1084 - Data assimilation has actually been performed for the radiation belts and electron ring current using VERB-3D and 4D and Van Allen Probes data. See https://agupubs.onlinelibrary.wiley.com/doi/10.1029/2020JA028208 and https://agupubs.onlinelibrary.wiley.com/doi/full/10.1029/2018SW002110.

Corrections incorporated in revised manuscript.

The paper then concludes with a discussion, summary, and some open questions.

 As I mentioned in the beginning discussion and summary are rewritten.
